# Health improvement framework for actionable treatment planning using a surrogate Bayesian model

Kazuki Nakamura [1,2], Ryosuke Kojima [2], Eiichiro Uchino [2], Koh Ono [3], Motoko Yanagita [4,5], Koichi Murashita[6], Ken Itoh [7], Shigeyuki Nakaji [8] & Yasushi Okuno [2✉]

Clinical decision-making regarding treatments based on personal characteristics leads to effective health improvements. Machine learning (ML) has been the primary concern of diagnosis support according to comprehensive patient information. A prominent issue is the development of objective treatment processes in clinical situations. This study proposes a framework to plan treatment processes in a data-driven manner. A key point of the framework is the evaluation of the actionability for personal health improvements by using a surrogate Bayesian model in addition to a high-performance nonlinear ML model. We first evaluate the framework from the viewpoint of its methodology using a synthetic dataset. Subsequently, the framework is applied to an actual health checkup dataset comprising data from 3132 participants, to lower systolic blood pressure and risk of chronic kidney disease at the individual level. We confirm that the computed treatment processes are actionable and consistent with clinical knowledge for improving these values. We also show that the improvement processes presented by the framework can be clinically informative. These results demonstrate that our framework can contribute toward decision-making in the medical field, providing clinicians with deeper insights.

[1] Research and Business Development Department, Kyowa Hakko Bio Co., Ltd., Tokyo, Japan. [2] Department of Biomedical Data Intelligence, Graduate School of Medicine, Kyoto University, Kyoto, Japan. [3] Department of Cardiovascular Medicine, Graduate School of Medicine, Kyoto University, Kyoto, Japan. [4] Department of Nephrology, Graduate School of Medicine, Kyoto University, Kyoto, Japan. [5] Institute for the Advanced Study of Human Biology, Kyoto University, Kyoto, Japan. [6] Center of Innovation Research Initiatives Organization, Hirosaki University, Hirosaki, Japan. [7] Department of Stress Response Science, Hirosaki University Graduate School of Medicine, Hirosaki, Japan. [8] Department of Social Health, Hirosaki University Graduate School of Medicine, Hirosaki, Japan. ✉email: okuno.yasushi.4c@kyoto-u.ac.jp

Medically appropriate and patient-acceptable decision making is beneficial in enhancing the quality of care[1–3]. Considerable evidence has been accumulated from many studies on the assessment of health statuses and risk profiles. Accordingly, standardized care has been provided in the form of guidelines. Conversely, the uniform application of standardized care is undesirable in real clinical situations whereby diversity exists in relation to individual preferences, feasibility, and acceptability[4]. Personalized medicine, also known as precision medicine, has become popular and presents new opportunities in clinical situations in recent years[5–7]. In addition to the understanding of individual, unique health-related factors, their consideration has been essential for shared decision making between patients and clinicians. Realistic and appropriate health improvement plans for the patient's health conditions have been regarded as a crucial component of shared decision making[8,9]. However, feasible planning largely depends on the empirical judgment of the clinicians in clinical situations, and decision making based on objective health improvement plans for both patients and clinicians is difficult. Major challenges remain in providing clinicians with tools to support clinical decision making in a data-driven manner[10,11].

Machine learning (ML) technology has been extensively used in the medical field, especially for diagnosis support and disease prediction based on comprehensive patient information[12–15]. Unlike interpretable techniques, such as classical statistical analysis and linear models, the black-box nature of highly predictive ML models, including ensemble learning and deep learning, is often a barrier to clinical decision-making applications[16]. Explainable artificial intelligence (XAI) has been receiving increasing attention recently in the field of ML[17]. XAI is a research field on techniques that explain black-box ML predictions, and it has been applied to medical ML models where interpretability is often required[18,19]. A successful application of XAI in the medical field is the identification of individual health-related factors that contribute to disease prediction using local interpretable model-agnostic explanations (LIME) and Shapley additive explanations (SHAP)[20–25]. These methods achieve both predictive performance and individual interpretation by using an additional individual model referred to as the surrogate model. The remaining important issue in clinical decision making is the development of personalized treatment plans for rational treatment[26]. However, these conventional methods merely explain the prediction reasons but cannot provide effective treatment processes. For example, for the prediction of hypertension, it is unclear what type of actions will improve effectively the individual blood pressure in a set of candidate actions related to blood test data and body composition despite the fact that we can understand important features in the prediction.

This study proposes a framework for planning an actionable path for personalized treatment based on the predictions of an ML model. A key idea of our framework is to use a hierarchical Bayesian model as a surrogate model of a specified ML model. We refer to this surrogate model as the stochastic surrogate model. Unlike the conventional surrogate model, the stochastic surrogate model based on the hierarchical Bayesian model enables the calculation of the probability of being the state of a given variable set. Our framework can evaluate the actionability of the treatment processes that was not considered by conventional methods[20,21], by computing a path probability with the use of the stochastic surrogate model. The combined use of the ML model and the stochastic surrogate model achieves both a high-prediction performance and actionability evaluation. The simultaneous computation of ML model prediction and actionability evaluation yields an actionable treatment process that leads to clinical applications to improve personal health.

This study also presents experiments conducted to address two different aspects. First, we evaluated our proposed framework from the viewpoint of the proposed methodology using a synthetic dataset. Subsequently, we assessed our framework on a clinical application using an actual health checkup dataset. In this experiment, we calculated personal health improvement paths and confirmed the consistency with clinical knowledge based on the assumption of two kinds of scenarios, wherein we aimed to lower blood pressure in individuals with systolic blood pressure (SBP) ≥ 140 mmHg and intervene individuals with a risk of chronic kidney disease (CKD), as examples for regression and classification problems, respectively. Also, we performed clinician assessments of the improvement process presented by the framework. To the best of our knowledge, this is the first study that made it possible to present effective improvement paths based on the ML model using an actual health checkup dataset.

## Results

**Path planning framework using surrogate Bayesian model.** This section describes the proposed framework (Fig. 1, Supplementary Fig. 1). Our framework consisted of three steps. Step 1: we built a prediction model using ML methods. Step 2: the surrogate model of the prediction model was constructed using hierarchical Bayesian modeling. Step 3: path planning was conducted to identify an actionable path for the treatment performed using the surrogate model. The actionability of the path was defined as the product of probabilities of taking a series of variable states on a specified path (detailed in "Methods"). Thus, the combined framework of the prediction model and the stochastic surrogate model achieves both high prediction performance and actionability evaluation, which leads to an actionable treatment process for personal health improvement.

In Step 1, a prediction model was built from a dataset expressed in the form of a table format, wherein columns consist of multiple explanatory variables and a response variable, and rows represent instances (Fig. 1a). The output of the constructed model would correspond to a clinician's assessment of individual health status or future predictions. For example, a regression model for blood pressure could be used to estimate the value of the response variable, i.e., blood pressure, from the value of explanatory variables, such as the body composition and blood test data. In our framework, we can use arbitrary ML algorithms, such as high-performance nonlinear algorithms, to construct the prediction model.

Based on the original explanatory variable values and the predicted values of the prediction model in Step 1, a stochastic surrogate model was constructed in Step 2 using hierarchical Bayesian modeling (Figs. 1b and 2). Items that cannot be easily measured in clinical situations, or future values, are available as the response variables given that the predicted values of the prediction model were applied in our framework. Clinically, this stochastic surrogate model represents a set of realistic health conditions for patients. The model was used to compute the probability of given values of explanatory and response variables to evaluate actionability in the next step. Note that this stochastic surrogate model represents the probability density for all possible states of variables.

In Step 3, a health-improvement treatment path was calculated for each instance using the stochastic surrogate model (Figs. 1c, d, and 3, as detailed in "Methods"), and would be equivalent to the consideration of an appropriate health improvement plan for individual patients in clinical situations. The explanatory variables of an instance, such as body composition and blood test data, were hypothetically changed to improve the response variable predicted using the prediction model. By using the

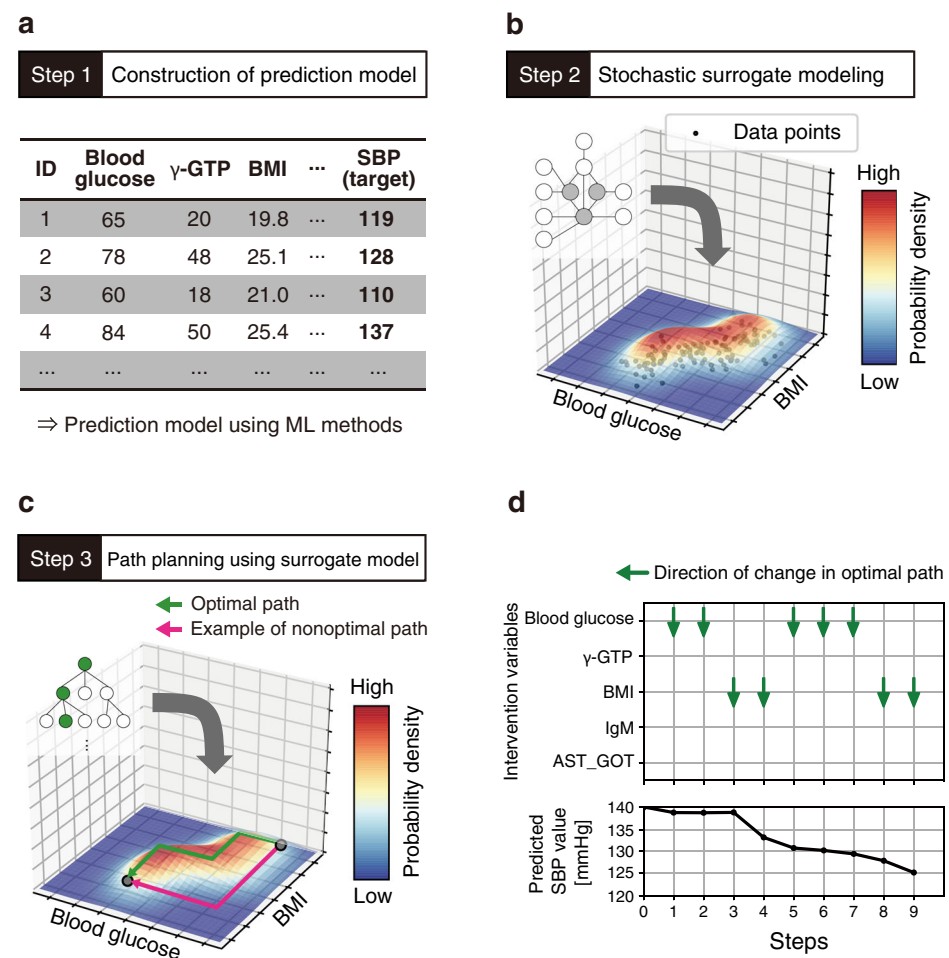

**Fig. 1 Schematic representation of the framework for planning actionable paths for treatment using hierarchical Bayesian modeling.** The framework consists of three steps. A schematic is given as an example in which a path is planned to improve the systolic blood pressure (SBP) owing to changes in blood data and body composition data. **a** Construction of a prediction model from the dataset. A variable SBP is set as the response variable in this case. **b** Construction of a stochastic surrogate model based on the original dataset and the predicted values of the prediction model. This figure shows a schematic representation of a two-variable space regarding blood glucose and body mass index (BMI). The heatmap and vertical axis represent the existence probability of data in the variable space, which is expressed by the stochastic surrogate model. **c, d** Actionable path planning is applied to improve the response variable. The path is represented as a set of multistep transitions on explanatory variables. In our framework, the optimal path (green line in (**c**)) is planned on the grid graph with high probabilities in the variable space based on the stochastic surrogate model. Conversely, the nonoptimal path (red line in (**c**)) could pass through nodes with low probability. γ-GTP gamma glutamyl transferase, IgM immunoglobulin M, AST_GOT aspartate transaminase.

stochastic surrogate model constructed in Step 2, we could calculate how easy it was to take the state of the combination of variables. The framework output the most actionable (optimal) path for a state that improved the prediction value, which was a sequence of the changed values with high probability in the surrogate model for the given ML model (detailed in "Methods"). Therefore, this model can avoid the nonoptimal health-improving paths (shown by the red line in Fig. 1c) where intermediate situations could be unrealistic.

Considering real applications, explanatory variables contain variables that cannot be changed by interventions such as age. The appropriate subset of variables for the application should be determined. The explanatory variables for setting the virtually changed values were called intervention variables in this study. In our path planning setting, the intervention variables were regarded as a grid graph, and a path was defined by connecting the grid points (nodes). We defined a probability of the node as the probability of taking the node calculated using the surrogate model. Based on a breadth-first search[27], we obtained the optimal

path to the destination node that achieves the most improved predictive value within the search iteration count $L$ (detailed in "Methods").

**Validation of framework on synthetic dataset**. We evaluated our framework with a synthetic dataset to confirm that improvement paths with high actionability could be planned. This synthetic dataset was generated from three, three-dimensional (3D) normal distributions (Supplementary Fig. 2). We built a regression model and subsequently constructed a stochastic surrogate model using hierarchical Bayesian modeling (Supplementary Fig. 3a, b). The lowest widely applicable Bayesian information criterion (WBIC) value[28] was obtained when the number of mixture components in the hierarchical Bayesian model was equal to two. Details of the setting related to this model are given in "Methods". Subsequently, we planned paths to decrease the value of the response variable using this stochastic surrogate model. All explanatory variables were selected for intervention variables, and planned paths were more actionable than the baseline path

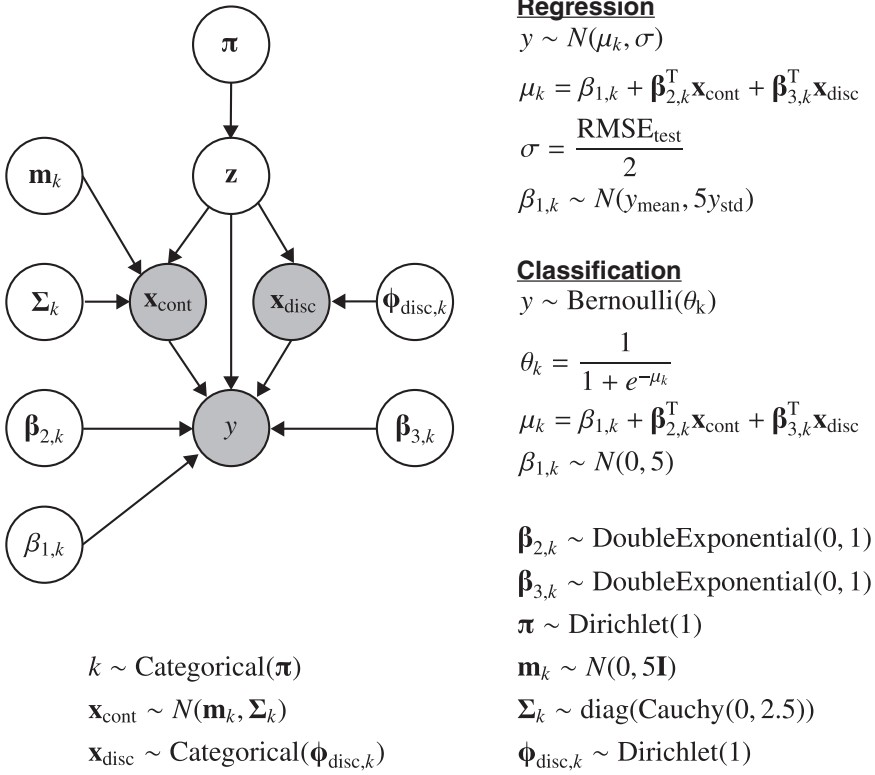

**Regression**

$$y \sim N(\mu_k, \sigma)$$

$$\mu_k = \beta_{1,k} + \boldsymbol{\beta}_{2,k}^{\mathrm{T}} \mathbf{x}_{\mathrm{cont}} + \boldsymbol{\beta}_{3,k}^{\mathrm{T}} \mathbf{x}_{\mathrm{disc}}$$

$$\sigma = \frac{\mathrm{RMSE}_{\mathrm{test}}}{2}$$

$$\beta_{1,k} \sim N(y_{\mathrm{mean}}, 5y_{\mathrm{std}})$$

**Classification**

$$y \sim \mathrm{Bernoulli}(\theta_k)$$

$$\theta_k = \frac{1}{1 + e^{-\mu_k}}$$

$$\mu_k = \beta_{1,k} + \boldsymbol{\beta}_{2,k}^{\mathrm{T}} \mathbf{x}_{\mathrm{cont}} + \boldsymbol{\beta}_{3,k}^{\mathrm{T}} \mathbf{x}_{\mathrm{disc}}$$

$$\beta_{1,k} \sim N(0, 5)$$

$$\boldsymbol{\beta}_{2,k} \sim \mathrm{DoubleExponential}(0, 1)$$

$$\boldsymbol{\beta}_{3,k} \sim \mathrm{DoubleExponential}(0, 1)$$

$$\boldsymbol{\pi} \sim \mathrm{Dirichlet}(1)$$

$$\mathbf{m}_k \sim N(0, 5\mathbf{I})$$

$$\boldsymbol{\Sigma}_k \sim \mathrm{diag}(\mathrm{Cauchy}(0, 2.5))$$

$$\boldsymbol{\phi}_{\mathrm{disc},k} \sim \mathrm{Dirichlet}(1)$$

$$k \sim \mathrm{Categorical}(\boldsymbol{\pi})$$

$$\mathbf{x}_{\mathrm{cont}} \sim N(\mathbf{m}_k, \boldsymbol{\Sigma}_k)$$

$$\mathbf{x}_{\mathrm{disc}} \sim \mathrm{Categorical}(\boldsymbol{\phi}_{\mathrm{disc},k})$$

**Fig. 2 Graphical model representation of stochastic surrogate model.** Nodes in the graphical model are represented as follows: $\mathbf{x}_{\mathrm{cont}}$, continuous explanatory variables; $\mathbf{x}_{\mathrm{disc}}$, discrete explanatory variables; $y$, response variable predicted by the ML model; $\mathbf{z}$, the parameter of the mixture components; and all the others, prior distributions. The formulation for $y$ differs between regression and classification tasks. $k$ represents each mixture component, and $\Sigma_k$ is a diagonal matrix with elements according to the Cauchy distribution. The symbol $\mathrm{RMSE}_{\mathrm{test}}$ in the equation represents a root-mean-squared error of the regression model, and $y_{\mathrm{mean}}$ and $y_{\mathrm{std}}$ represent the mean and standard deviation values of the predicted response variable, respectively.

---

**Algorithm 1** Path search algorithm

**function** PATH_SEARCH
1: *initial_node* ← data of instance
2: *initial_node.cost* ← 0
3: All other *node.cost* ← ∞
4: *initial_node.visited* ← False
5: All other *node.visited* ← False
6: *current_node* ← *initial_node*
7: **for** $i = 1$ to $L$ **do**
8: *neighbors* ← list of *node* adjacent to *current_node* in grid graph
9: **for** *neighbor_node* in *neighbors* **do**
10: *cost* ← *current_node.cost* + *neighbor_node.neg_log_prob*
 (Negative logarithm of node probability)
11: **if** *neighbor_node.cost* > *cost* **then**
12: *neighbor_node.cost* ← *cost*
13: **end if**
14: **end for**
15: *current_node.visited* ← True
16: *current_node* ← *node* which is not *visited* and has smallest *cost*
17: **end for**
18: *destination_node* ← *node* with the best response variable
19: **return** *destination_node*

**Fig. 3** The intervention variable space was regarded as a grid graph, and the grid points (nodes) were connected to plan a path. The nodal probability was calculated using the surrogate model. The actionability was defined as the product of nodal probabilities on a specified path. The most actionable (optimal) path for each node was calculated, and the output path was the optimal path to the node with the most improved predictive value within the search iteration count, $L$. Pseudocode of path search algorithm.

---

(Supplementary Fig. 3c). This baseline method for the baseline path is defined in "Methods". The planned paths of two randomly selected instances are shown in Fig. 4. We successfully demonstrated that our framework could plan paths to improve response variable values with high probabilities in our framework. Another

experiment on a more complicated five-dimensional dataset also demonstrated successful path planning (Supplementary Notes). These results showed that the planned paths with high probabilities were discovered rather than the naïve straight path that connected the initial node to the destination node.

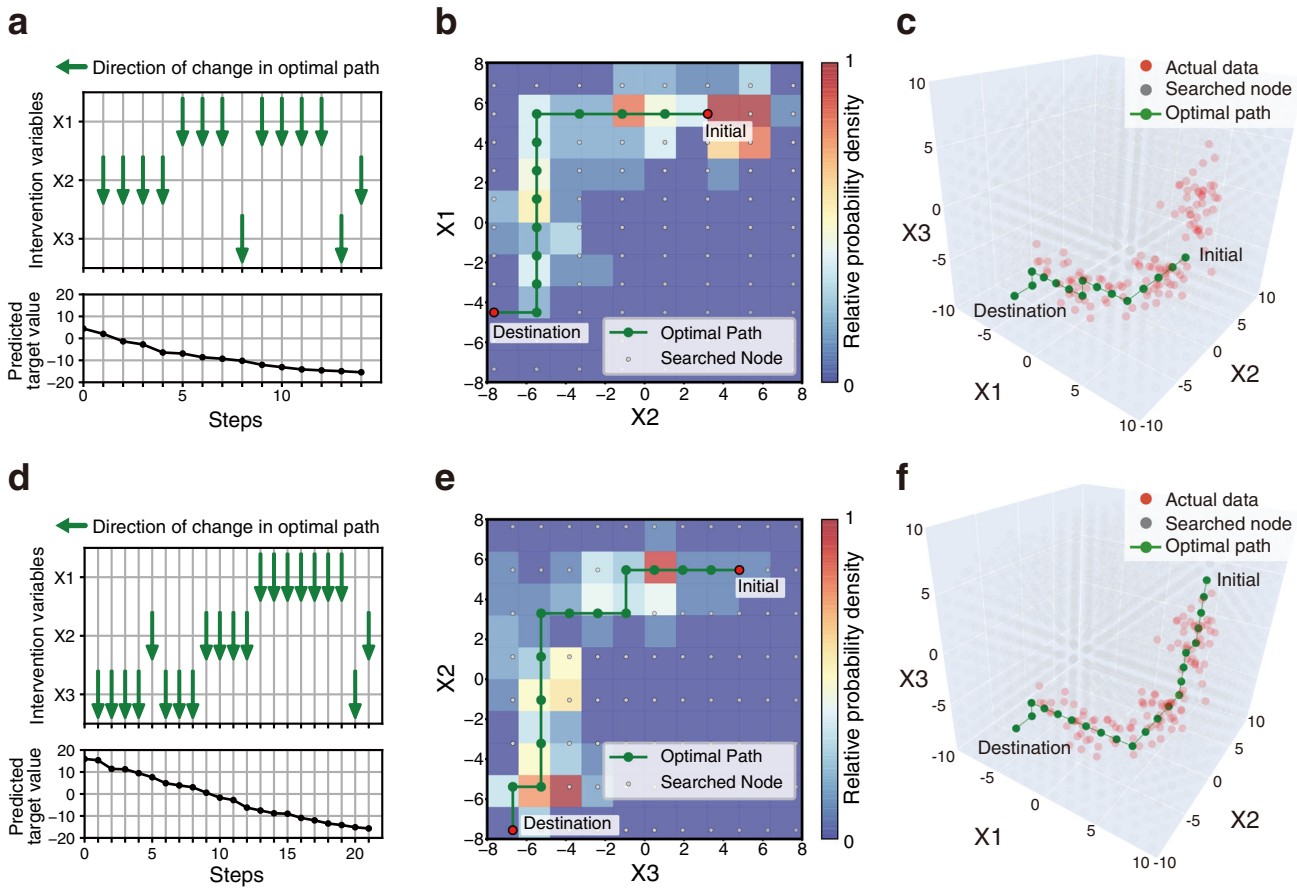

**Fig. 4 Examples of actionable paths planned on synthetic dataset.** The optimal paths for improving the response variable predicted by the ML model are represented for randomly selected two examples: instance A (**a–c**) and instance B (**d–f**). **a, d** The orders of changes in the explanatory variables in the optimal path and the accompanying changes in the predicted values. In the transition steps, the upward or downward arrow represents a unit increase or decrease in the explanatory variable, respectively. **b, e** Two-dimensional (2D) plots of the path. The 2D plots are shown regarding the selected two variables: X1 and X2 (**b**), and X2 and X3 (**e**). In the heatmaps, the probability density of the actual data, normalized by the panel with the maximum number of data, is expressed. **c, f** Three-dimensional (3D) plots of the path.

| Table 1 Subject characteristics during first-time participation. | |
| --- | --- |
| **Participant characteristics** | **(n = 3132)** |
| Age (years) | 51.3 ± 16.0 |
| BMI (kg/m$^2$) | 23.0 ± 3.5 |
| SBP (mmHg) | 125.9 ± 19.1 |
| DBP (mmHg) | 74.8 ± 11.9 |
| eGFR (mL/min/1.73 m$^2$) | 82.1 ± 16.5 |
| Sex | |
| Male | 1234 (39.4%) |
| Female | 1898 (60.6%) |
| History of hypertension | |
| No history | 2446 (78.1%) |
| Undertreatment | 650 (20.8%) |
| Past history | 36 (1.1%) |
| *BMI* body mass index, *SBP* systolic blood pressure, *DBP* diastolic blood pressure, *eGFR* estimated glomerular filtration rate. | |

**Application on actual health checkup dataset.** We used the Iwaki health promotion project (IHPP) dataset, an actual health checkup dataset, to demonstrate that the proposed framework can plan actionable paths for treatment. The IHPP has acquired (on an annual basis) a wide range of health checkup data that describe the molecular biology, physiology, biochemistry, lifestyle, and socio-environment of participants. We considered two kinds of scenarios to lower SBP and risks for CKD, whereby planned paths could be interpreted from a clinical perspective. CKD risk was defined as estimated glomerular filtration rate (eGFR) <60 mL/min/1.73 m$^2$. Table 1 shows an overview of the IHPP dataset. Because the dataset comprised more than 2000 measurement items, we reduced the explanatory variables before we built prediction models in our framework. We excluded diastolic blood pressure (DBP) and limb blood pressures from the explanatory variables for SBP prediction and creatinine for CKD risk prediction. Our framework assumes the intervention in explanatory variables to improve response variables. Hence, actions such as intervening in blood pressure (such as DBP) to lower SBP are not reasonable. Furthermore, ambiguous items, such as answers to questionnaires and items with ≥25% missing values, were excluded. Subsequently, XGBoost-based[29] recursive feature elimination (RFE)[30] was performed to reduce the explanatory variables with the training data (Supplementary Data 1 and Supplementary Table 1). We applied one-hot encoding for categorical variables and replaced the missing values with the median of the training data for simplicity. RFE was performed with five-fold cross-validation split by participants, and the explanatory variables were reduced to 25, which had little impact on predictive scores on validation data (Supplementary Fig. 4). In the following sections, applications of our framework on the regression task for SBP and classification task for CKD risk are described.

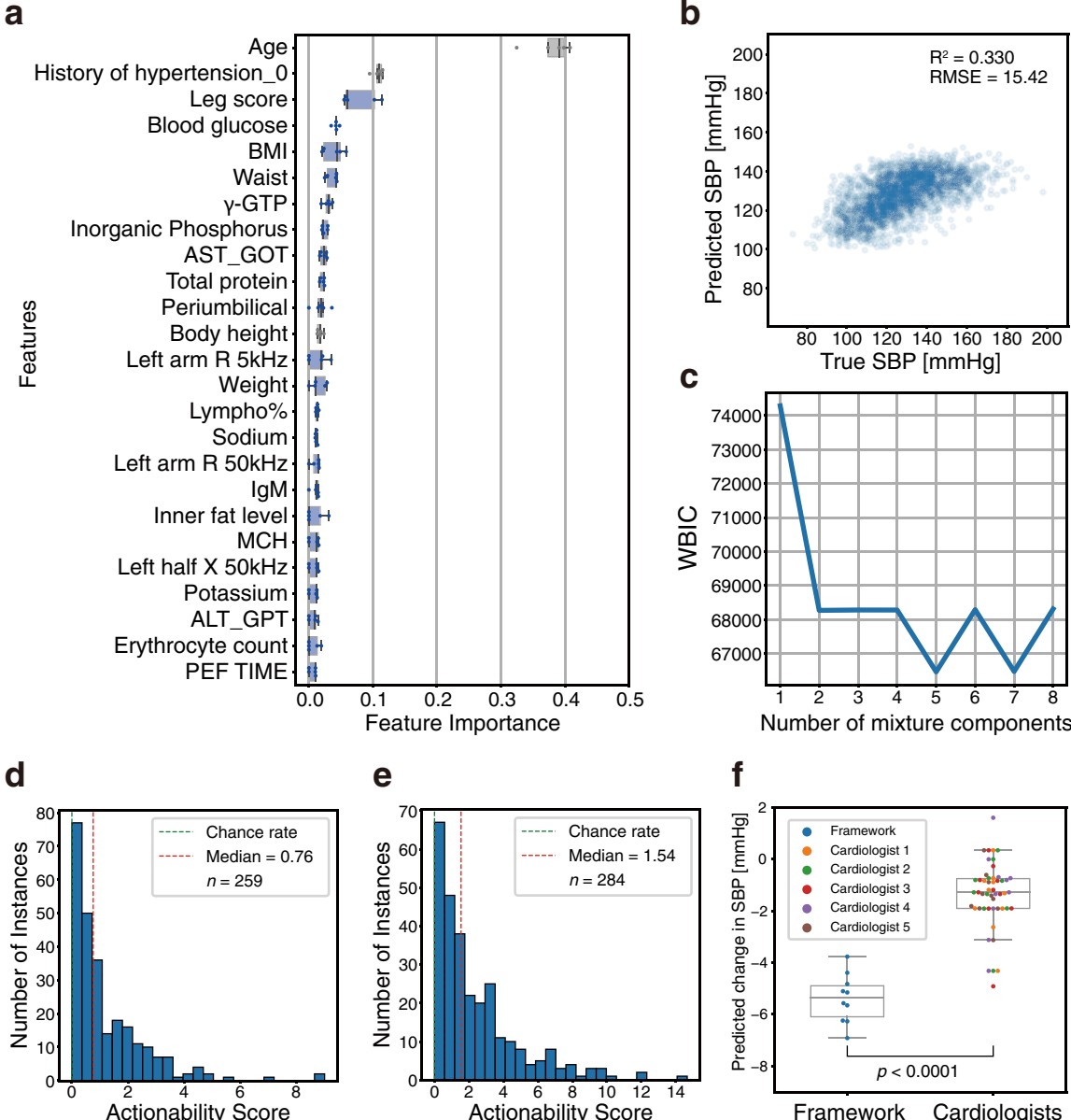

**Fig. 5 Application of proposed framework on systolic blood pressure (SBP) regression task using the Iwaki health promotion project (IHPP) dataset.**
**a** Feature importance: these 25 features were selected by recursive feature elimination (RFE) to predict the SBP. RFE was performed with fivefold cross-validation, and the feature importance when 25 variables remained is shown for each fold ($n = 5$). The plot color represents the following: gray: variables which cannot be intervened, and blue: intervenable variables. Details of features are described in Supplementary Data 1. **b** Plot for prediction vs. true response variable. **c** Widely applicable Bayesian information criterion (WBIC) values of stochastic surrogate models with 1–8 mixture components. **d**, **e** Histogram of actionability scores with intervention variables based on data-driven selection (**d**) or hypothesis-driven selection (**e**) at different instances. An actionability score of zero indicates that the actionability of the optimal path is equivalent to that of the baseline path. **f** Comparison of predicted SBP reduction between framework-proposed paths and cardiologist-selected paths. Health improvement paths constructed based on hypothesis-driven intervention variables using our framework were compared with the cardiologist-selected paths among framework-proposed paths and random paths. Each cardiologist evaluated the same randomly selected instances ($n = 10$). Statistical significance was calculated using the Welch's $t$-test (two-sided). In box-plot, center line represents median; box limits, upper and lower quartiles; whiskers, 1.5× interquartile range.

**Application of framework on SBP regression task**. We applied our framework to the SBP regression task. Important features selected by RFE comprised items related to hypertension, such as age, body composition (leg score, body mass index [BMI], and waist), blood glucose, gamma glutamyl transferase (γ-GTP), and serum sodium[31–40] (Fig. 5a, the details of the variables are described in Supplementary Data 1). Therefore, the selected explanatory variables were considered to be reasonable for SBP predictive models from the clinical perspective.

Following our framework, a regression model was built after the replacement of missing values with the use of multiple imputations. This is a more precise imputation method for missing values. To estimate multiple imputations, Bayesian ridge and random forest[41] were used for continuous and discrete variables, respectively. Consequently, the regression model yielded a root-mean-squared error (RMSE) equal to 15.42 and an R-squared value equal to 0.330 (Fig. 5b). Subsequently, hierarchical Bayesian modeling was performed to construct the

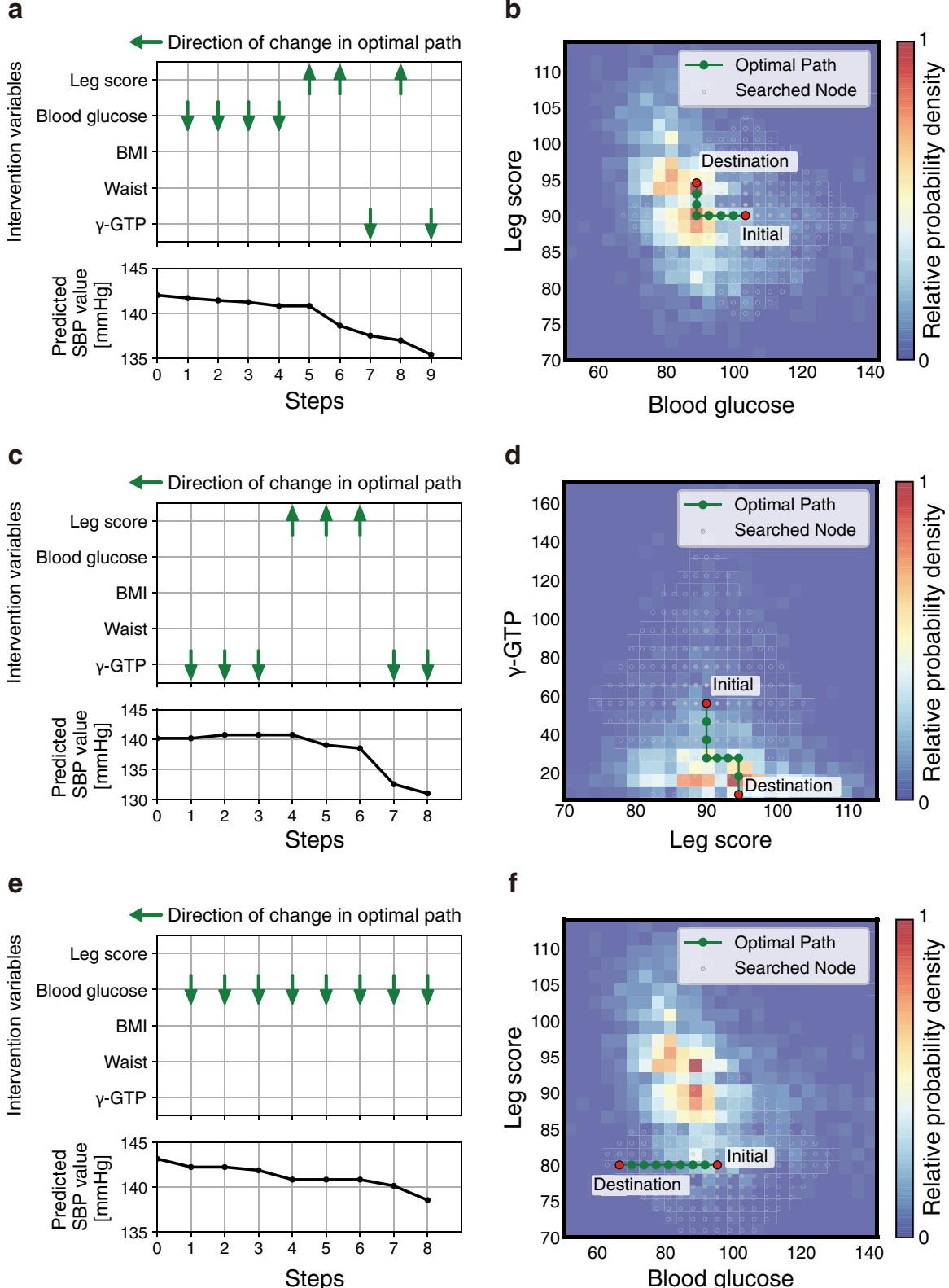

stochastic surrogate model based on multiple imputed RFE-selected features and predicted SBP. The lowest WBIC value was obtained when the number of mixture components was five (Fig. 5c). Path planning was performed with this stochastic surrogate model. Regarding the intervention variables in path planning, we examined two patterns for selection methods: data-driven selection and hypothesis-driven selection.

First, the top-five variables that could be intervened were selected based on feature importance in a data-driven selection case: leg score, blood glucose, BMI, waist, and γ-GTP (Fig. 5a). Variables that could be changed by direct or indirect means, unlike such as body height and medical history, were handled as intervenable. The unit cell size of the grid was set to 0.2 σ for each explanatory variable, where σ represents the standard deviation of

**Fig. 6 Examples of personal actionable paths for treatment with intervention variables based on data-driven selection in systolic blood pressure (SBP) regression task.** The optimal paths for improving the response variable predicted by the ML model are represented for randomly selected three examples: instance 1 (**a**) and (**b**), instance 2 (**c**) and (**d**), and instance 3 (**e**) and (**f**). **a**, **c**, **e** The orders of changes in the explanatory variables in the optimal path and the accompanying changes in the predicted values. In the transition steps, the upward or downward arrow represents a unit increase or decrease in the explanatory variable, respectively. **b**, **d**, **f** 2D plots of the path. The 2D plots are shown regarding the two influential variables in the optimal path: blood glucose and leg score (**b**), leg score and gamma glutamyl transferase (γ-GTP) (**d**), and blood glucose and leg score (**f**). In the heatmaps, the probability density of the actual data, normalized by the panel with the maximum number of data, is expressed. 3D plots of the path are shown in Supplementary Fig. 5. BMI body mass index.

the training data. Assuming a scenario wherein the task is to lower the SBP in participants with higher values, relevant instances were selected according to the following criteria: predictive SBP ≥ 140 mmHg[42] and no missing values in the intervention variables. The number of applicable instances was 259. We executed the path-search algorithm with $L = 20,000$ for each instance and acquired a path to the node with the lowest predictive SBP value in count $L$. For quantitative evaluations, we introduce the actionability score for each instance that indicates how actionable the planned path was compared with the baseline path (detailed in "Methods"). The histogram of the actionability score is shown in Fig. 5d. The actionability scores were greater than zero, that is, planned paths were more actionable than baseline paths in 227/259 instances, and the median was 0.76. This result suggested that even if the response variable value after improvement was the same, the path planned by our framework had a higher actionability than the baseline path in most cases. The paths of the three randomly selected instances are shown in Fig. 6 and Supplementary Fig. 5. The actual data scatters also support the fact that the path was planned to pass through areas with high-nodal probabilities. In instance 1, it was shown that the path that improved the values of the variables in the following order was more effective: blood glucose, leg score, and γ-GTP (Fig. 6a, b). These variables are related to each other, and the path in which multiple variables fluctuated to improve the blood pressure was reasonable[43–45]. In instance 2, the path that improved the values in the following order was more valid: γ-GTP, leg score, and again γ-GTP (Fig. 6c, d). The SBP values predicted by the regression model increased temporarily compared with the original SBP value. In instance 3, the optimal path was planned only based on the improvement of one explanatory variable, namely, the blood glucose (Fig. 6e, f). The actionability score yielded a value of zero because the optimal path was identical to the baseline path in such cases.

Second, we also planned paths with another intervention variables set considering a more realistic hypothesis-driven selection. Assuming actual instruction, the following four variables that would fluctuate with the clinical guideline–recommended treatments[42] were selected by cardiologists as intervention variables: blood glucose, BMI, γ-GTP, and serum sodium (Supplementary Table 2). Instance selection method and other settings in path planning were the same as in the case of data-driven selection. The number of applicable instances was 284. Actionability scores were greater than zero in 265/284 instances, and the median was 1.54 (Fig. 5e). Furthermore, we performed two steps of cardiologist assessments of the improvement paths proposed by the framework. First, we compared the reduction in the predictive SBP of the framework-proposed improvement paths with the cardiologist interventions. As an alternative expression for the cardiologist's intervention, the most suitable path was selected from ten improvement paths, which included a framework-proposed path and nine randomly intervened paths (Supplementary Fig. 6, detailed in "Methods"). According to the guideline[42], the direction of improvement was set to decrease for all intervention variables when generating random paths.

Intervention-induced changes in predictive SBP were not displayed to the cardiologists at the time of selection. The framework-proposed paths exhibited a significant decrease in predictive SBP compared to the paths selected by the cardiologists (Fig. 5f). Subsequently, the utility of the planned paths was evaluated. The cardiologists assessed the practicality and informativeness of a framework-proposed path and random path, respectively (Supplementary Fig. 7). The ratio of the paths that the cardiologists evaluated as informative in the framework-proposed paths was $0.42 \pm 0.30$ (mean ± standard deviation) (Supplementary Table 3). Details of the cardiologists' assessment of suggestive individual instances are provided in the Supplementary Notes.

**Application of framework on CKD risk classification task.** We also applied our framework to the CKD risk classification task. Instead of improving the predicted value in the regression task, we aimed to reduce the predicted probability of the prediction model in the classification task using our framework. According to the guideline for CKD, the eGFR cutoff value for the classification task was set to 60 mL/min/1.73 m²[46]. RFE-selected features comprised items related to renal function, such as age, uric acid, blood urea nitrogen (BUN), triglyceride, anemia-related factors (hemoglobin [Hb], hematocrit [Ht], and erythrocyte count), and blood pressure–related factor (right ankle DBP [RADIA])[47–52] (Fig. 7a). Hence, the selected explanatory variables were considered to be reasonable for CKD risk prediction.

Following our framework, a classification model and stochastic surrogate model were constructed after the replacement of missing values with the use of the precise imputation method, the multiple imputation method. The classification model yielded an area under the curve (AUC) equal to 0.844 (Fig. 7b). Subsequently, we constructed a stochastic surrogate model based on hierarchical Bayesian modeling. The lowest WBIC was obtained when the number of mixture components was five (Fig. 7c). Using this stochastic surrogate model, we performed path planning in two sets of intervention variables: data-driven selection and hypothesis-driven selection. Assuming a scenario wherein the task is to intervene the participants with decreased renal functions, relevant instances were selected according to the following criteria: predictive class of eGFR <60 and no missing values in the intervention variables. We executed the path-search algorithm with $L = 20,000$ for each instance and acquired a path to the node with the lowest predicted probabilities of eGFR <60 in count $L$. Other settings in path planning were the same as in the case of SBP.

First, the top-five variables that could be intervened were selected based on feature importance in a data-driven selection case: uric acid, BUN, Hb, immunoglobulin A (IgA), and triglyceride (Fig. 7a). The number of applicable instances was 33. Actionability scores were greater than zero in 29/33 instances, and the median was 3.51 (Fig. 7d). The paths of the three randomly selected instances are shown in Fig. 8 and Supplementary Fig. 8. In instance 4, it is shown that the path of improvement in the order of BUN and Hb is effective and succeeds in reducing

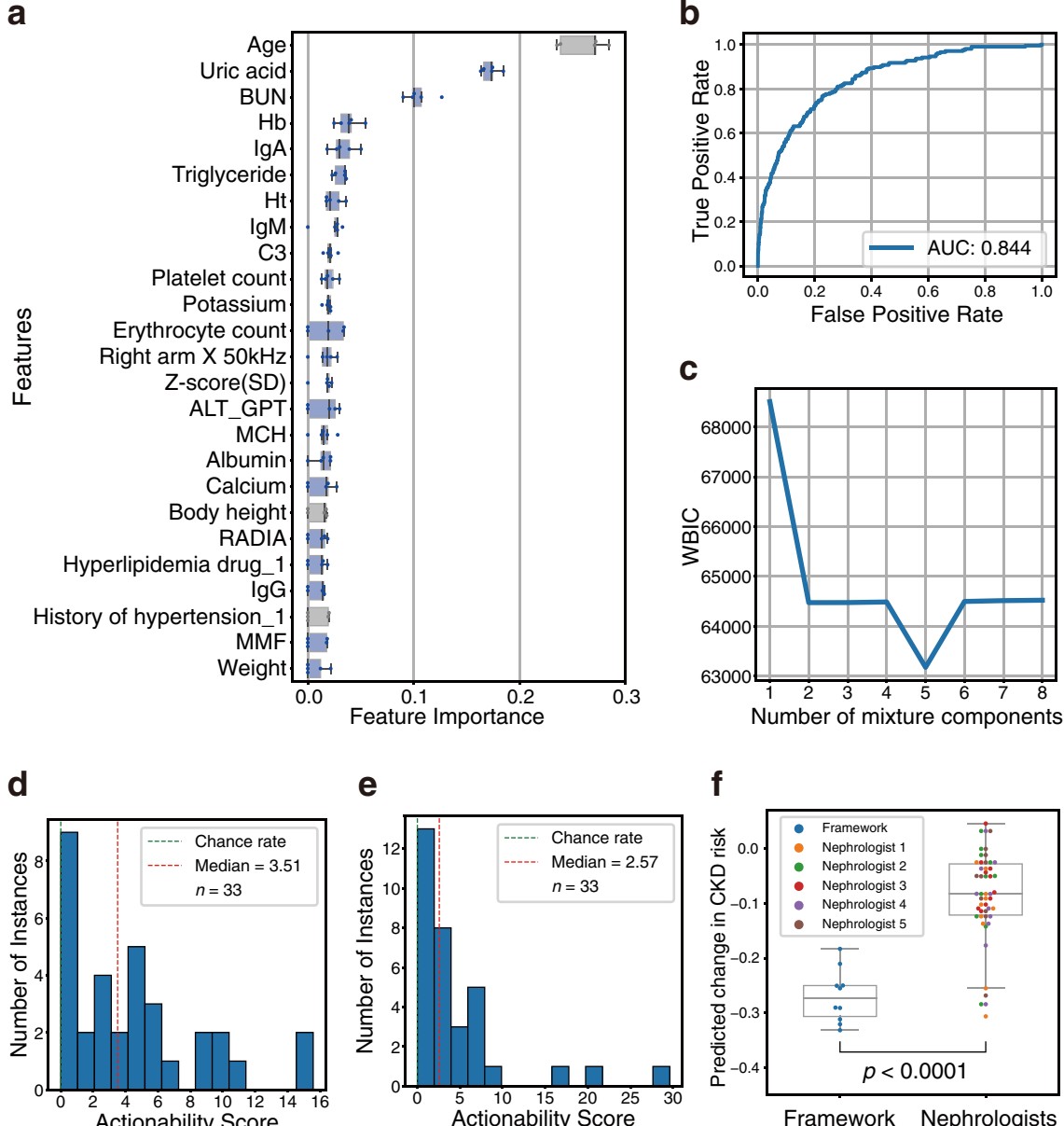

**Fig. 7 Application of proposed framework on chronic kidney disease (CKD) risk classification task using the Iwaki health promotion project (IHPP) dataset. a** Feature importance: these 25 features were selected by recursive feature elimination (RFE) to predict the estimated glomerular filtration rate (eGFR) category. RFE was performed with five-fold cross-validation, and the feature importance when 25 variables remained is shown for each fold ($n = 5$). The plot color represents the following: gray: variables which cannot be intervened, and blue: intervenable variables. Details of features are described in Supplementary Data 1. **b** Classification model score. AUC area under the curve. **c** Widely applicable Bayesian information criterion (WBIC) values of stochastic surrogate models with 1–8 mixture components. **d, e** Histogram of actionability scores with intervention variables based on data-driven selection (**d**) or hypothesis-driven selection (**e**) at different instances. An actionability score of zero indicates that the actionability of the optimal path is equivalent to that of the baseline path. **f** Comparison of predicted CKD risk reduction between framework-proposed paths and nephrologist-selected paths. Health improvement paths constructed based on hypothesis-driven intervention variables using our framework were compared with the nephrologist-selected paths among framework-proposed paths and random paths. Each nephrologist evaluated the same randomly selected instances ($n = 10$). Statistical significance was calculated using the Welch's $t$-test (two-sided). In box-plot, center line represents median; box limits, upper and lower quartiles; whiskers, 1.5× interquartile range.

the prediction probability of eGFR <60. In both instances 5 and 6, though the optimal path mainly consisted of improvements in the order of uric acid and BUN, the amount of the total change in each variable was different.

Second, we also planned paths with another intervention variables set based on hypothesis-driven selection. Four variables that would fluctuate with the clinical guideline–recommended treatments[46] were selected by nephrologists: triglyceride, RADIA,

weight and Hb (Supplementary Table 4). The number of applicable instances was 33. Actionability scores were greater than zero in 32/33 instances, and the median was 2.57 (Fig. 7e). Furthermore, we performed nephrologist assessments of the paths proposed by the framework. First, the reductions in CKD risk, defined as the probability of eGFR <60 given by the prediction model, were compared between framework-proposed paths and nephrologist interventions. The most suitable path was

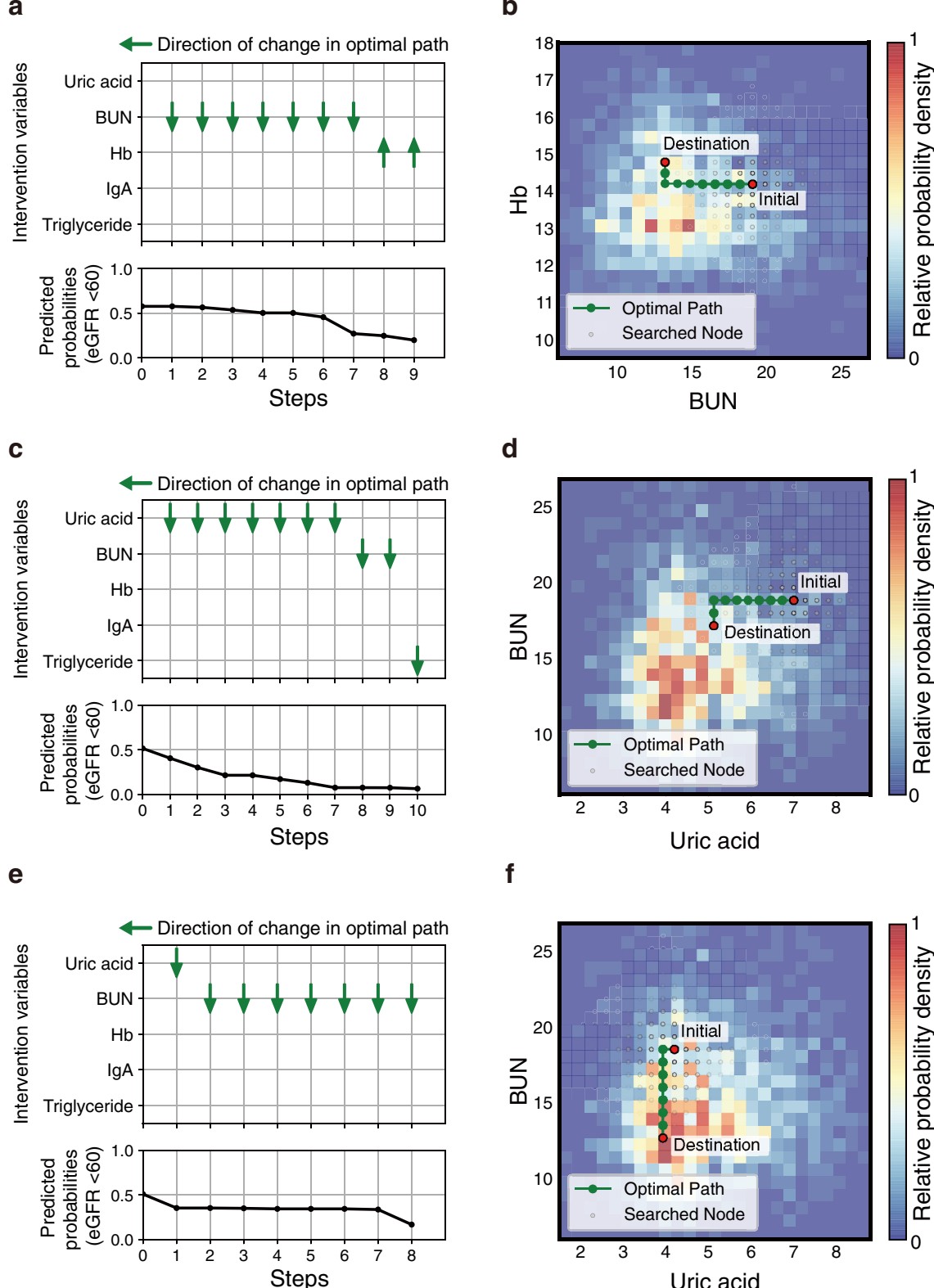

selected from a framework-proposed path and nine random paths by the nephrologists. Based on the guideline[46], the direction of improvement when generating random paths was set as follows: increase for Hb and decrease for triglyceride, RADIA, and weight. The framework-proposed paths exhibited a significant decrease in predictive CKD risk compared to the paths selected by the

nephrologists (Fig. 7f). Subsequently, the utility of the planned paths and random paths were evaluated by the nephrologists. The ratio of paths that nephrologists evaluated as informative in framework-proposed paths was 0.52 ± 0.26 (Supplementary Table 3). Details of the nephrologists' evaluation of suggestive individual instances are provided in the Supplementary Notes.

**Fig. 8 Examples of personal actionable paths for treatment with intervention variables based on data-driven selection in chronic kidney disease (CKD) risk classification task.** The optimal paths for improving the response variable predicted by the ML model are represented for randomly selected three examples: instance 4 (**a**) and (**b**), instance 5 (**c**) and (**d**), and instance 6 (**e**) and (**f**). **a**, **c**, **e** The orders of changes in the explanatory variables in the optimal path and the accompanying changes in the predicted values. In the transition steps, the upward or downward arrow represents a unit increase or decrease in the explanatory variable, respectively. **b**, **d**, **f** 2D plots of the path. The 2D plots are shown regarding the two influential variables in the optimal path: hemoglobin (Hb) and blood urea nitrogen (BUN) (**b**), and BUN and uric acid (**d**) and (**f**). In the heatmaps, the probability density of the actual data, normalized by the panel with the maximum number of data, is expressed. 3D plots of the path are shown in Supplementary Fig. 8. eGFR estimated glomerular filtration rate, IgA immunoglobulin A.

## Discussion

In this study, we proposed a framework for planning paths to improve the prediction values of ML models. We demonstrated that the proposed framework could plan paths through nodes with high probabilities using the synthetic datasets. Furthermore, our proposed framework was capable of planning actionable paths to improve the predicted SBP and CKD risk values in the actual health dataset. The results of applying our framework for the eGFR regression task and the hypertension risk classification task described in the Supplementary Notes also show the successful path planning using our framework. Though conventional surrogate models applied in XAI methods, such as LIME and SHAP, are useful for identifying individual factors that contribute to prediction, they cannot provide the probability of taking variable states. In our framework, the construction of a stochastic surrogate model based on hierarchical Bayesian modeling enabled the estimation of joint probability densities for virtually changed variables and actionable path planning. Our framework suggests realistic concrete treatment processes that are actionable on humans for personal health improvement.

As shown in Figs. 6 and 8, our framework could visually present concrete improvement paths at the individual level. The direction of change in intervention variables was consistent with conventional clinical knowledge for intervention in patients with hypertension[33–39,42] and CKD[46,48,50]. For example, high blood glucose levels have been reported to be a risk factor of hypertension, and reduction of triglyceride is an appropriate intervention for patients with impaired renal function. Furthermore, the planned paths consisted of a sequence of changes in intervention variables and could be translated into actual clinical treatments using correspondence tables between the variables and clinical guideline–recommended treatments (Supplementary Tables 2 and 4). In this way, our framework could plan health improvement paths using the intervention variables that would change as a result of clinicians' guidance based on clinical guidelines. Even if the guidelines to be referred to would be changed depending on the geographical region and clinical domain in practical use, the framework-proposed paths could be associated with the clinicians' guidance by using the appropriate guidelines. As a result of clinicians' assessment, the paths planned using our framework exhibited significant improvement in predictive response variables compared to the paths selected by clinicians (Figs. 5f and 7f). Also, the improvement paths presented by our framework were informative for clinicians to some extent (Supplementary Table 3). However, the ratio of paths that clinicians evaluated as practical was not high (Supplementary Table 3). In this study, though we performed an objective path planning without considering clinical constraints, practical adjustments would be needed (detailed in the Supplementary Notes). Tool-assisted goal-settings are expected to help time-constrained clinicians and contribute to better health improvement in patients[53]. Our framework can provide clinicians with understandable and informative health improvement plans based on patient health data and given intervention variables.

Accordingly, this can be suitable for patient–clinician collaborative decision making on health interventions.

In the remaining part of this section, we provide methodological considerations of the proposed framework. Although the experiments were performed under specific settings, our framework has generality in terms of its methodology owing to three aspects. First, we used XGBoost[29] to build the prediction models to be explained in the main manuscript. Because our framework operates in a model-agnostic manner, other high-performance ML models can be used. The Supplementary Notes provide the results of applying the framework to random forest and support vector machine (Supplementary Figs. 9–29 and Supplementary Tables 5–6). In the synthetic datasets, similar paths were planned for the same instances regardless of applied ML algorithms (Fig. 4 and Supplementary Figs. 9–15). However, in the actual dataset, the paths differed in the same instances depending on the ML algorithms (Figs. 6, 8, and Supplementary Figs. 16–29). The actual dataset was more complicated than the synthetic datasets, which resulted in the construction of the prediction models with different properties. Our framework aimed to provide an optimal path to the destination node that would improve the predictions of the ML model. Therefore, the destination node itself was changed when the properties of the ML model were different, which resulted in large differences in the planned paths. Second, although we assumed a mixture distribution of normal or categorical distributions for the explanatory variables in the hierarchical Bayesian modeling (Fig. 2), distributions can be selected according to the data. This is expected to be applied to some extent to medical data, which is often accompanied by a significant amount of noise and missing values. Finally, we selected intervention variables based on the data-driven or hypothesis-driven manner in the path planning with the health checkup data. Also, the unit cell size in planning was set to $0.2\ \sigma$ of the data distribution. These selection methods and values can be flexibly changed according to the application or the patient's request and environment.

Our objective in path planning was to obtain a path to the best-predicted value with the set conditions of the number of iterations $L$. In real situations, application-dependent or clinical constraints would exist. A typical case is that the values of explanatory and/or response variables should be less/more than the reference values in all nodes along the path. For example, there were cases where the predicted SBP values were temporarily increased from the original value in our experiment (Fig. 6c). This would be better avoided in clinical situations. By slightly changing the path search condition to exclude undesirable nodes, our framework can be applied to these cases. In addition, there would be use cases where the target value of the response variable or intervention variable should be determined by clinicians (Supplementary Notes). Our framework can be applied by modifying the termination conditions of the search to reach the target value.

From the perspective of expanding the proposed framework, the high-computational cost when many intervention variables exist or when calculating a long-term path for treatment needs to be considered. Subject to our experimental conditions

($L = 20{,}000$ and five intervention variables), ~10 min were required for path planning per instance. Under the condition of more intervention variables, the number of steps of the planned paths decreased, and the calculation time became longer even in the same iteration count, $L$ (Supplementary Figs. 30,31). A more practical path planning can be expected by combining our framework with techniques for finding intervention points, such as counterfactual explanations[54–58]. Counterfactual explanations usually present intervention goal values of the explanatory variables for changing the response variable without considering the intervention process. Our framework can plan actionable paths to the intervention goal values decided by counterfactual explanations. For this case, a more efficient path planning algorithm, such as the A* search algorithm[59], can be applied in our framework. Similar to our actionable path planning approach, some studies have been conducted in recent years to obtain intervention points with minimum costs[58]. These approaches enable fast searches by assuming linearity. Our framework is more suitable for use with sophisticated nonlinear ML models, i.e., in cases where linearity is difficult to assume, such as those pertaining to medical checkup data.

Our study has some limitations. First, the health checkup dataset used in this study was obtained from a single area and had a small sample size. This could have contributed to the low-prediction score of the SBP regression model (Fig. 5b). Although the dataset problem does not impair the validity of the proposed framework as a methodology, we performed supplementary experiments that apply the framework on the public datasets to support the framework's validity (Supplementary Notes). Besides, though we selected variables that could be directly or indirectly modifiable as intervention variables because of the nature of health checkup data, it is preferable to build a prediction model with more variables that can be directly intervened. Also, because of the limitation of the observational study data, the stochastic surrogate model only calculated the joint probability density of a set of variables and did not explicitly express the intervention effect considering the causality[60]. By estimating the causal effect of each intervention variable on the response variable through more controlled studies related to individual applications, it may be possible to construct a stochastic surrogate model considering the causality and to present health-improvement paths that are more consistent with the clinician's consideration. Second, in path planning, we set a commonly used and easy-to-interpret grid graph, which resulted in the constraint of changing one variable at a time. Because our framework intervened in explanatory variables stepwise, the fluctuation on the correlated variables was observed as a pattern in which multiple variables alternated. In most cases, multiple variables fluctuated in the optimal paths (Supplementary Fig. 32). In actual application, graphs other than the grid can also be applied according to requirements. Finally, we encountered a problem when we verified the clinical effectiveness of the paths planned by using our framework. Planned paths in the synthetic dataset indicated that our framework would perform correctly. Also, we evaluated that the change directions of intervention variables in some paths were consistent with clinical knowledge in the clinical application. The clinician evaluation results suggested that while the framework would be promising, there are points to be adjusted for its practical application. It is necessary to verify the effectiveness of the paths through a prospective cohort study to suit the real-world applications of our framework.

In conclusion, we proposed a framework to plan actionable health improvement processes at the individual level. Using the synthetic dataset, we proved that our framework could plan actionable paths through the nodes with high probabilities. Furthermore, we successfully demonstrated that health-improving

paths planned for lowering blood pressure and improving CKD risk based on the application of our framework to the actual health checkup dataset were actionable and consistent with clinical knowledge. Our framework can present reasonable and personalized health improvement plans based on ML model predictions in a wide range of situations that is expected to contribute to decision making in the medical field. Further studies should focus on the prospective clinical validation of actionable paths planned by using the framework proposed herein. Our framework may provide clinicians with deeper insights by proposing definite and actionable treatment paths through the use of the ML model.

## Methods

**Synthetic 3D dataset**. We generated a simple 3D dataset to verify whether our framework can plan paths by transiting the nodes with high probabilities in the variable space to improve the predicted response values of the prediction model. The dataset was generated from three, 3D normal distributions to ensure that straight paths were not always actionable (Supplementary Fig. 2). Each distribution generated 200 data points that consisted of $x_1$, $x_2$, and $x_3$. The response variables were set to the sum of $x_1$, $x_2$, and $x_3$ with Gaussian noise ($\sigma = 2$). The dataset consisted of 600 data points and randomly split into training data (80%) and test data (20%).

**IHPP dataset**. To evaluate our framework, we used the IHPP dataset. In this study, we considered the use cases to plan actionable paths to improve the SBP for a regression task and CKD risk for a classification task. CKD was defined by eGFR value, which was calculated using the equation for Japanese population[61]. According to the guideline risk classifications, the eGFR cutoff value to 60 mL/min/1.73 m²[46]. The IHPP has annually acquired a wide range of health checkup data that comprise the molecular biology, physiology, biochemistry, lifestyle, and the socio-environmental aspects of residents of the Iwaki district, Hirosaki City, Aomori Prefecture, Japan. In this study, we targeted 12,803 health checkup instances for 13 years (2005–2017). Given the existence of cases where the same person participated in multiple years, the number of unique participants was 3132 (Table 1). The dataset was randomly split into training (80%) and test data (20%) by participants (Supplementary Table 1). This study was approved by the Ethics Committee of Hirosaki University School of Medicine (approval number: 2019-009) and was conducted according to the recommendations of the Declaration of Helsinki. All participants provided written informed consent.

**Construction of prediction models**. In the experiments, XGBoost[29], which is based on a gradient-boosting decision tree algorithm, was used to create the prediction models. In general, XGBoost is a high-performance, nonlinear model. We also examined prediction models based on other nonlinear ML algorithms in supplementary experiments: random forest and support vector machine. The hyperparameters of the model were determined by fivefold cross-validation of the training data. For preprocessing, continuous explanatory variables were standardized by the mean and standard deviation. In addition to this preprocessing, techniques such as other data-dependent preprocessing were applied. These data-dependent preprocessing is described in the results section.

**Stochastic surrogate model with hierarchical Bayesian modeling**. The graphical model representation of the surrogate model constructed in this study is shown in Fig. 2. In this case, $\mathbf{x}_{\text{cont}}$ represents continuous explanatory variables, such as body composition and blood test data, $\mathbf{x}_{\text{disc}}$ represents discrete explanatory variables, such as sex, and $y$ represents a response variable, such as the blood pressure value. We denote the measured explanatory values as $\mathbf{x}$ and the predictions by the prediction model as $y$. $\mathbf{z}$ is the parameter of the mixture components, and $k$ represents each mixture component. The data generative process on a regression task is formulated as follows

$$k \sim \text{Categorical}(\pi) \tag{1}$$

$$\mathbf{x}_{\text{cont}} \sim N(\mathbf{m}_k, \Sigma_k) \tag{2}$$

$$\mathbf{x}_{\text{disc}} \sim \text{Categorical}(\phi_{\text{disc},k}) \tag{3}$$

$$y \sim N(\mu_k, \sigma) \tag{4}$$

$$\mu_k = \beta_{1,k} + \beta_{2,k}^T \mathbf{x}_{\text{cont}} + \beta_{3,k}^T \mathbf{x}_{\text{disc}} \tag{5}$$

$$\sigma = \frac{\text{RMSE}_{\text{test}}}{2} \tag{6}$$

where $\text{RMSE}_{\text{test}}$ is an RMSE of the regression model to adjust to the

nonstandardized $y$ scale. The priors for other hyperparameters in Eqs. (1)–(5) are defined as follows

$$\beta_{1,k} \sim N\left(y_{mean}, 5y_{std}\right) \quad (7)$$

$$\beta_{2,k} \sim \text{DoubleExponential}(0, 1) \quad (8)$$

$$\beta_{3,k} \sim \text{DoubleExponential}(0, 1) \quad (9)$$

$$\pi \sim \text{Dirichlet}(\mathbf{1}) \quad (10)$$

$$\mathbf{m}_k \sim N(0, 5\mathbf{I}) \quad (11)$$

$$\Sigma_k \sim \text{diag}\left(\text{Cauchy}(0, 2.5)\right) \quad (12)$$

$$\phi_{disc,k} \sim \text{Dirichlet}(\mathbf{1}) \quad (13)$$

where $y_{mean}$ and $y_{std}$ represent the mean and standard deviation values of the predicted response variable to adjust to the nonstandardized $y$ scale, respectively. $\Sigma_k$ is a diagonal matrix with elements according to the Cauchy distribution. In a classification task, $y$ is a binary variable and Eqs. (4)–(7) are changed to the following formula accordingly:

$$y \sim \text{Bernoulli}\left(\theta_k\right) \quad (14)$$

$$\theta_k = \frac{1}{1 + e^{-\mu_k}} \quad (15)$$

$$\mu_k = \beta_{1,k} + \beta_{2,k}^T \mathbf{x}_{cont} + \beta_{3,k}^T \mathbf{x}_{disc} \quad (16)$$

$$\beta_{1,k} \sim N(0, 5) \quad (17)$$

We used PyStan[62] to estimate the parameters using the Markov chain Monte Carlo algorithm (iteration = 1500, warm-up = 500). For model selection, the WBIC was calculated for each model[28]. For a stable training, instances with values outside the $3\sigma$ range calculated using the training data in the continuous explanatory variables were excluded as outliers.

The priors were selected from noninformative prior distributions or weakly informative prior distributions. The relationships between prior distribution hyperparameters and WBIC are shown in Supplementary Figs. 33–36. Furthermore, we set 1–8 as the range of mixture components from the viewpoint of calculation costs and supplementary experimental results (Supplementary Figs. 37–40). The supplemental results show that the planned paths almost unchanged under a superabundant number of mixture components.

All analyses were carried out using custom software written in Python. Open-source Python packages (pandas, numpy, scipy, xgboost[29], scikit-learn, matplotlib, and pystan[62]) were used in our framework.

**Path planning using a stochastic surrogate model**. We calculated a path for the treatment for each instance based on a breadth-first search algorithm[27]. The intervention variable space was regarded as a grid graph, and the grid points (nodes) were connected to plan a path. We defined the probability of the node as the probability of taking the node calculated using the surrogate model. Furthermore, the actionability was defined as the product of nodal probabilities on a specified path. The purpose of this algorithm was used to obtain the most actionable (optimal) path for each node. The output path was the optimal path to the node with the most improved predictive value within the search iteration count, $L$. From the computational perspective, we used the negative logarithm of actionability as a path cost to minimize, which is mathematically synonymous with maximizing actionability.

The pseudocode of this algorithm is shown in Fig. 3. In lines 2–5, the cost and visited state (the optimal path was searched or not) of nodes were initialized. The search started from the initial node in line 6. We obtained a list of nodes adjacent to the currently selected node in the grid graph in line 8 of the pseudocode. Subsequently, the costs for these nodes were updated in lines 10–12. The following node was selected in line 16. After reaching the predetermined count $L$, the optimal path to the node with the best ML model prediction value was selected as the planned path in line 18. If multiple nodes with the same predictions existed, the path with the minimum cost was selected.

**Actionability score**. To evaluate the actionability of the paths, we defined the actionability score expressed by the following equation: log(optimal path actionability)—log(baseline path actionability), where the optimal path actionability is the actionability of the path planned using our framework (Supplementary Fig. 41). The baseline path actionability is the geometric mean of ten actionabilities of paths that connect both ends of the optimal path by the shortest procedure in a random manner. The actionability score indicates the log scale of the ratio of the actionability of the optimal path to the actionability of the baseline path. Hence, the higher the actionability score, the better the optimal path is planned through nodes with higher probabilities than the baseline path. When the actionability score is zero, the optimal path has the same actionability as the baseline path.

**Clinician assessment of health-improvement paths**. We conducted two steps of assessments. First, we compared the predicted value reductions between the framework-proposed paths and clinicians' interventions. We randomly selected items from the intervention variables and generated random paths in which these items were intervened in a random order on the grid graph in the appropriate direction based on the treatment policy of clinical guidelines. The number of interventions, that is, steps, in the random path was set to be the same as the path planned using the framework in each instance. Nine patterns of random improvement paths were created for each of ten randomly selected instances. For each instance, clinicians selected an improvement path considered to be the closest to their treatment policy among ten blinded improvement paths, which comprised of a path planned by our framework and nine random paths (Supplementary Fig. 6). When the clinicians made a selection, only the order of interventions and the initial values of measurement items of the instance was presented, and the transition of the predictive response variable was not displayed. Five board-certified members of the Japanese Circulation Society assessed the paths in the SBP regression task and five board-certified nephrologists of the Japanese Society of Nephrology in the CKD risk classification task. Regarding the changes in the predictive response variables of the paths proposed by the framework and the paths selected by the clinicians, a two-tailed Welch's $t$-test was used to determine a significant difference between the two groups.

Second, we assessed the utility of the improvement paths presented by our framework. The clinicians were provided with the initial values of measurement items of the instance, as well as two blinded paths: the framework-proposed path and the random path (Supplementary Fig. 7). The order of interventions along with the accompanied transition of the predictive response variable and the path projected onto 2D heatmaps, which exhibited the actual data distribution, for each pair of intervention variables were presented. The clinicians assessed the practicality and informativeness of paths respectively for ten instances.

**Reporting summary**. Further information on research design is available in the Nature Research Reporting Summary linked to this article.

## Data availability

The synthetic datasets can be generated from the code in the repository provided in the Code availability section. The datasets used in the Supplementary Information are open available on UCI machine learning repository[63] and Trevor Hastie's Software page at https://web.stanford.edu/~hastie/Papers/LARS/. The health checkup data used in this study were collected in the Iwaki Health Promotion Project (IHPP) and transferred to a secure data center with restricted access controls in a de-identified format. The de-identified data are available from Hirosaki University School of Medicine (contact via e-mail: coi@hirosaki-u.ac.jp) for academic research purposes only and for researchers who meet the criteria for access to the data. Researchers need to be approved by the research ethics review committees of both the Hirosaki University School of Medicine and their affiliation. Three months are to be expected for the access request to be approved. All other data in this study are included in this article or are available from the corresponding author upon reasonable request.

## Code availability

We provide the custom scripts to perform our framework at (https://github.com/clinfo/actionable_path_planning) with an assigned (https://doi.org/10.5281/zenodo.4706186).

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

## Acknowledgements

This work was supported by JST, Center of Innovation Program (JPMJCE1302), and Kyowa Hakko Bio Co., Ltd. The author, Kazuki Nakamura, thanks Kazushi Shoji, Miho Komatsu, Takashi Ishida, and Yuko Shimanami, employees of Kyowa Hakko Bio Co., Ltd., for their generous support. The authors thank Minoru Sakuragi, M.D. for helpful comments on our paper. The authors would like to thank cardiologists Kazuki Matsushita, M.D., Masahiro Kimura, M.D., Ph.D., Osamu Baba, M.D., Ph.D., Sawa Miyagawa, M.D., and Shuhei Tsuji M.D. for their cooperation in clinical assessment. The authors appreciate nephrologists Daisuke Takada, M.D., Naoya Toriu, M.D., Noriaki

Sato, M.D., Shusuke Hiragi, M.D., MBA, Ph.D., and Takuya Ishimura, M.D. for their cooperation in clinical assessment.

## Competing interests

Kazuki Nakamura is an employee of Kyowa Hakko Bio Co., Ltd. All other authors declare no competing interests.
