## [Peer Review File · Nature Communications]

Reviewers' Comments:

Reviewer #1:

Remarks to the Author:

Nakamura et al. present a paper proposing a framework for predicting and planning treatment decisions using a surrogate model of a machine-learned model, employ a surrogate model to evaluate the "actionability" of treatment processes. They claim to employ the surrogate model to obtain an actionable, optimal path for health improvement by modeling a patient's outcomes along a health trajectory.

After testing on a theoretical dataset, they employ a dataset from a Japanese Cohort study (IHPP) which conducted longitudinal inferences on a set of patients with the goal of modeling and improving systolic blood pressure measurements. They reduced 2000 explanatory features to 25 with XGBoost recursive feature elimination. With these 25 variables they build MI regression models and performed path planning with the surrogate model, selecting 5 (or 25?) values which could be intervened upon.

In general the authors' work is well considered and novel, but upon close inspection the clinical and scientific utility of this tool is not currently meaningful for several reasons as explained below. The authors could improve this tool to make it more meaningful.

Major

- The methodology of the paper seems mostly careful and well considered. The authors use good techniques for feature selection and imputation. I have no major concerns about the theoretical or technical aspects of the paper.

- My major concern about this paper is that it is unclear to me the clinical impact or real-world meaning of the discovered paths to be taken. I am not sure that the "paths" to be taken are actually actionable. There are no drugs suitable for patients with hypertension which could be employed to alter GGT, for example. Furthermore, many possible interventions would alter many of the variables simultaneously—for example, weight loss. Why should we try to intervene on only one variable at one time? This is not explored well in the paper. Other variables with high importance seem to not be modifiable. What is "Left arm R 5Khz", for example?

- It is unclear to me what the "actionability score" actually represents. They define it as the log (optimal path) - log (baseline path) scores.

- In general, I recommend that the authors rethink their variable inclusion strategy. They should focus on selecting variables in a hypothesis-driven fashion instead of a data-driven fashion, and select variables which may be truly modifiable in a patient dataset. They could then explain how their tool could be used to make actual clinical recommendations. For example, a fruitful area in hypertension may be the sequence of drugs in hypertensive patients.

- Selection of reducing blood pressure was done by SBP above mean + (1-sigma). This does not reflect clinical guidelines, which are based upon cutoffs for hypertension (e.g. SBP >140mmHg). Can the authors justify their choice? It does not seem to be correct to select patients to modify in this fashion.

Minor

- Reduced explanatory features to 25 with XGBoost recursive feature elimination. Why was 25 selected? What criteria were used?

- Explanatory variables related to SBP were excluded. It is unclear what the authors mean when they state this. Please provide more information. Why do the authors believe this is justified? It

seems like these are precisely the variables which should be retained.

- Correlated features: Why were BMI and Waist size both selected? They should be highly correlated features and most feature-selection algorithms would remove one of them.
- Please provide a list of all variables considered at all steps in the process.
- In 2020, having code available "upon reasonable request" for a mathematical model or tool is unacceptable. The authors should release their code for this method under appropriate license and make it available online for inspection.
- There is no such thing as "cured" hypertension (Table 1)

Reviewer #2:

Remarks to the Author:

Advances in machine learning have meant that many machine learning models that produce impressive predictions/results are considered "black boxes," as there is no easy way to explain how the models arrived at the results that they produced. This is especially problematic when such models are utilized in clinical medicine, where physicians often want to understand the "reasoning" that produced a particular result before acting on the result. The authors present a surrogate model based on hierarchical Bayesian models that can explain a nonlinear black box model and plan treatment or action paths.

Work on explainable AI is becoming more common in the biomedical domain and is often based on using decision trees as surrogate models that clinicians can more easily understand. A major strength of the current work is the use of an actual clinical dataset related to systolic blood pressure management. While the work presented is interesting, there are many missing details that would improve it.

Since the paper title states that it is presenting a health improvement framework, more examples related to health (as opposed to synthetic data) should be presented in the main paper (not in a supplement). For each medical problem presented, the authors should make clear how many features/predictors and samples were used in training the original nonlinear model and the surrogate, how missing data is handled in the original and in developing the surrogate and the rationale behind the methods chosen for handling missing data, and how well the original nonlinear model predicts the outcome of interest for both classification and regression problems.

While a prospective study might be the best way to assess the effectiveness of treatment planning (path planning), an intermediate evaluation might involve asking several expert clinicians in the domain to assess the treatment plans produced by the surrogate model for their perceived utility.

As it currently stands, the paper makes claims about the potential utility of the methods presented for medical decision making but does not adequately substantiate these claims.

It would help to cite more of the recent literature on explainable/intelligible models in medicine, such as:

Caruana R, Lou Y, Gehrke J, Koch P, Sturm M, Elhadad N. Intelligible models for healthcare: predicting pneumonia risk and hospital 30-day readmission. In: Proceedings of the 21st ACM SIGKDD international conference on knowledge discovery and data mining; 2015.

Lenert MC, Matheny ME, Walsh CG. Prognostic models will be victims of their own success, unless. *J Am Med Inform Assoc.* 2019;26(12):1645–50.

Zhang A, Teng L, Alterovitz G. An explainable machine learning platform for pyrazinamide resistance prediction and genetic feature identification of *Mycobacterium tuberculosis*. *J Am Med Inform Assoc.* 2020; November 20.

Review

Manuscript NCOMMS-20-45495 entitled “Health improvement framework for planning actionable treatment process using surrogate Bayesian model”

Summary

The authors embark on solving an everlasting problem in healthcare: personalised actionable treatment regimes. They propose a framework for treatment planning taking advantage of the popularity and flexibility of machine learning models coupled with the interpretability of more classical statistical approaches. Although the effort is certainly noteworthy, I believe more work is needed towards that goal. Overall, the authors need to provide stronger evidence for their claims.

First, the paper needs some fine-tuning in terms of presentation, coherence and concision. It will make it much easier to read and appreciate. My main comments follow:

Comments

- Since the intention (at least, how I perceived it) is to use the proposed framework to make actionable decision in clinical practice, more rigorous experiments are needed. These include:
 - more extensive simulations using realistic data generating models. Even though simulations such as the one presented in the paper are useful to get intuition about the framework, more realistic data scenarios need to be created. For instance, the diagonal covariance matrices (Supplementary Figure 2) are unrealistic in real-life applications. Most, if not all datasets, exhibit some correlation between the explanatory variables. Also, the dimensionality of the simulations need to be higher than 3.
 - investigation of the impact of misspecification of the Bayesian surrogate model. The normality assumption for the covariates (eq 1) is not very realistic. For instance, blood pressure does not have the real line as support. Extensive simulations under different data generating models can provide indication of the robustness of the surrogate model.
 - one or two more real-life applications related to medicine. The authors need to consider examples with more intervention variables as these are the rule rather than the exception in clinical applications. This will allow to showcase how the search algorithm performs in such settings. (Since the algorithm seems to be making local moves I have some doubts whether it performs well in higher dimensions.)
- The disadvantage of the surrogate model. The authors claim that the Bayesian surrogate model allows to calculate the probability of counterfactual values (Results section, 4th paragraph). This statement needs justification. It is a very strong statement given the method uses observation data. To draw causal conclusions from observational data additional assumptions need to be made (see Chapter 3, Hernán MA, Robins JM, 2020).

Causal Inference: What If.). All the surrogate model does is estimate the joint densities of the explanatory variables. If some values are not seen in the dataset due to biases arising from the observational nature of the data, then they will be assigned low posterior probability and the path planning optimisation will avoid these regions of the space.

- Actionability is not formally defined in any part of the paper? The only reference close to a definition is “From the computational perspective, we used the negative logarithm of actionability, defined as the product of node probabilities on a path, as a cost of the path”, which raises further questions such as how the “cost of the path” is defined/evaluated? Since “actionability” is a core component of the method it needs to be formally defined.
- In the introduction the authors refer to other similar ML approaches arising from the field of XAI. It will be useful to outline the differences (advantages and disadvantages) of the field compared to the proposed framework and maybe discuss further in the Discussion section. For example, since both LIME and SHAP use a surrogate model the main contribution of the proposed methodology seems to be the path planning algorithm. If that is the case, can we use the surrogate model from LIME or SHAP followed by the proposed path planning optimisation routine? If yes, then the proposed method should be evaluated compared to LIME/SHAP or other similar approaches. If no, mention why.
- About the saerch algorithm. The breadth-first search algorithm needs to be briefly explained or at least offer a reference for the reader who wants to know more. In addition, I have a couple of reservations on the implementation. The authors state “this algorithm was used to obtain the most actionable path to the node that achieved the most improved predictive value within the search iteration count, L ”. This raises the issue that if I run the algorithm for $L + 1$ iterations I may come up with a different answer. This seems undesirable. Some form of convergence to an optimum needs to be established. Otherwise, there is no notion of an “optimal health-improvement treatment path” (Results, section, 4th paragraph). Also, what happens if the algorithm gets stuck in a node? Under normal circumstances the algorithm could be re-initialised from different values. But, here the initial node is fixed since it corresponds to the observed covariate values of a given patient. Lastly, the presentation of the pseudocode needs more work: what are the *currentnode.cost*, *negihbor.node.neg.log.prob* etc? I did not find a definition in the entire document. These quantities are crucial for someone who wants to implement the algorithm, so they should be clearly defined.
- Sensitivity analysis. The authors should evaluate the sensitivity of the results to a number of choices. First, what are implications if a different ML regression model is used? Even though XGBoost is fine, the authors need to show whether the path planning changes under different models. This is because different models lead to different predictions which may have (big) impact to all further downstream analysis. Second, please evaluate the sensitivity of the results on the choice of priors in the surrogate model and motivate their choice. Third, discuss/motivate the choice for the range of mixture components, why 1-8?

Reviewers' comments to Author & Reply to reviewers

To Reviewers

We appreciate the time and effort the reviewers have dedicated to providing insightful feedback on ways to strengthen our paper. We scrutinized our manuscript and noticed that Fig. 4a and 4d were reversed between the instances. We apologize for this oversight. This has been corrected in the revised manuscript.

To Reviewer #1

Thank you very much for your useful comments and suggestions regarding our manuscript, which have helped us significantly improve the paper.

Remarks to the Author:

Nakamura et al. present a paper proposing a framework for predicting and planning treatment decisions using a surrogate model of a machine-learned model, employ a surrogate model to evaluate the "actionability" of treatment processes. They claim to employ the surrogate model to obtain an actionable, optimal path for health improvement by modeling a patient's outcomes along a health trajectory.

After testing on a theoretical dataset, they employ a dataset from a Japanese Cohort study (IHPP) which conducted longitudinal inferences on a set of patients with the goal of modeling and improving systolic blood pressure measurements. They reduced 2000 explanatory features to 25 with XGBoost recursive feature elimination. With these 25 variables they build MI regression models and performed path planning with the surrogate model, selecting 5 (or 25?) values which could be intervened upon.

In general the authors' work is well considered and novel, but upon close inspection the clinical and scientific utility of this tool is not currently meaningful for several reasons as explained below. The authors could improve this tool to make it more meaningful.

Major

- The methodology of the paper seems mostly careful and well considered. The authors use good techniques for feature selection and imputation. I have no major concerns about the theoretical

or technical aspects of the paper.

Comment (1) My major concern about this paper is that it is unclear to me the clinical impact or real-world meaning of the discovered paths to be taken. I am not sure that the "paths" to be taken are actually actionable. There are no drugs suitable for patients with hypertension which could be employed to alter GGT, for example.

Answer: We thank the reviewer for this pertinent comment. In this study, we assumed improving intervention variables not only by direct means such as drugs but also by indirect means such as lifestyle-related improvements. In normal clinical situations, instructions such as lifestyle changes before medicine use would be tried. We agree with the reviewer's comment that "actually actionable" path planning should be taken. It seems preferable to select variables that can be directly intervened as intervention variables. Because the health data used in this study was health checkup data, there was a limitation that it was hard to construct a predictive model using only explanatory variables that could be directly intervened. Hence, we have added the following text as one of the limitations of this study:

Addition: "Besides, though we selected variables that could be directly or indirectly modifiable as intervention variables because of the nature of health checkup data, it is preferable to build a prediction model with more variables that can be directly intervened." (p. 15, 6th paragraph in the Discussion section)

Moreover, to overcome the limitation of using the health checkup data, we have added a table to translate our intervention variables into actual treatments. Supplementary Table 3 is a correspondence table between the variables and clinical guideline–recommended treatments for lowering blood pressure. For example, the altering γ -GTP mentioned in the reviewer's comments was thought to correspond with alcohol restriction. If a path in the order of blood glucose reduction, γ -GTP reduction, and sodium reduction is proposed, it corresponds to intervention guidance in the order of dietary pattern change, alcohol restriction, and salt intake restriction according to the guideline–recommended treatments.

We also performed an additional experiment to plan paths with intervention variables pre-selected by cardiologists. For example, four variables that would fluctuate with the clinical guideline–recommended treatments were selected by the cardiologists as intervention variables: blood glucose, BMI, γ -GTP, and serum sodium. The results of this additional experiment have been added as the 4th paragraph starting with the sentence "Second, we also planned paths with..." in the subheading "Application of framework on SBP regression task" in the Results section (p. 9–10). Figure 5e shows

that our framework could present improvement paths based on changing the intervention variable corresponding to guideline–recommended treatments. For example, Supplementary Fig. 41a–c means that while focusing mainly on dietary pattern change and exercise, it is preferable to limit salt intake and alcohol in the intermediate process.

As shown above, combining with Supplementary Table 3, our framework could plan the “actually actionable” paths that the reviewer mentioned. Since Supplementary Table 3 is based on Japanese guidelines for hypertension, different guidelines should be referred in case of the application to non-Japanese and other diseases. Therefore, actual intervention should be performed based on the guidelines according to the geographical region and clinical domain, when applying our framework in the actual clinical situations. We have added the following text in the Discussion section:

Addition: “Furthermore, the planned paths consisted of a sequence of changes in intervention variables and could be translated into actual clinical treatments using correspondence tables between the variables and clinical guideline–recommended treatments (Supplementary Table 3 and 5). In this way, our framework could plan health improvement paths using the intervention variables that would change as a result of clinicians' guidance based on clinical guidelines. Even if the guidelines to be referred to would be changed depending on the geographical region and clinical domain in practical use, the framework-proposed paths could be associated with the clinicians' guidance by using the appropriate guidelines.” (p. 12, 2nd paragraph in the Discussion section)

Comment (2) **Futhermore, many possible interventions would alter manyt of the variables simultaneously—for example, weight loss. Why should we try to intervene on only one variable at one time? This is not explored well in the paper.**

Answer: Thank you for your useful comment on our paper. As the reviewer mentioned, many possible interventions cause simultaneous changes in multiple variables. In response to the reviewer's comment, we performed an additional analysis on the number of the intervention variables that fluctuated in the paths (Supplementary Fig. 32). In most instances, multiple variables fluctuated in the optimal path. In this study, we used the grid graph for path planning because it is easy to interpret. The use of the grid graph imposed the constraint of changing one variable at one time in the intervention. Because the intervention is represented as a multistep variable change in the improvement paths, variables that simultaneously alter are considered to be pseudo-represented as patterns in which multiple variables fluctuate in the paths (Figs. 6a and c). Also, graphs other than the grid can be applied in our framework according to actual requirements. Therefore, we have added the following text as one of the limitations of this study:

Addition: “Second, in path planning, we set a commonly used and easy-to-interpret grid graph, which resulted in the constraint of changing one variable at a time. Because our framework intervened in explanatory variables stepwise, the fluctuation on the correlated variables was observed as a pattern in which multiple variables alternated. In most cases, multiple variables fluctuated in the optimal paths (Supplementary Fig. 32). In actual application, graphs other than the grid can also be applied according to requirements.” (p. 15, 6th paragraph in the Discussion section)

Comment (3) Other variables with high importance seem to not be modifiable. Waht is "Left arm R 5Khz", for example?

Answer: Thank you for your careful comment. There was insufficient explanation in the manuscript about the criteria for modifiable/intervenable. In this study, variables that could be changed by direct or indirect means, unlike such as body height and medical history, were considered to be modifiable/intervenable. Therefore, we have added the following text:

Addition: “Variables that could be changed by direct or indirect means, unlike such as body height and medical history, were handled as intervenable.” (p. 8, 3rd paragraph in the subheading “Application of framework on SBP regression task” in the Results section)

"Left arm R 5kHz", which was mentioned in the reviewer's comment, is a parameter related to the impedance measured by the body composition meter and reflects the body composition of muscle and fat. Hence, this item is considered to change due to exercise and improvement of food habits. We have added more detailed descriptions of all items to Supplementary Table 1.

Comment (4) It is unclear to me what the "actionability score" actually represents. They define it as the log (optimal path) - log (baseline path) scores.

Answer: Thank you for your constructive comment on our paper. In this study, the actionability of a path is defined as the product of nodal probabilities on the path, whereas the actionability score is defined as $\log(\text{optimal path actionability}) - \log(\text{baseline path actionability})$. The actionability score indicates how well the path presented using the framework was planned through nodes with high probabilities based on the actual distribution. To clarify the explanation, we have added a schematic explanation as Supplementary Fig. 39 and have modified the description in the main text as follows:

Before: “To evaluate the actionability of the paths,... The actionability score indicated the actionability of the optimal path compared with that of baseline paths.”

↓

After: “The actionability of a path was defined as the product of nodal probabilities on the path as described in the previous section. To evaluate the actionability of the paths,... The actionability score indicates the log scale of the ratio of the actionability of the optimal path to the actionability of the baseline path. Hence, the higher the actionability score, the better the optimal path is planned through nodes with higher probabilities than the baseline path.” (p. 19, 1st paragraph in the subheading “Actionability score” in the Methods section)

Comment (5) In general, I recommend that the authors rethink their variable inclusion strategy. They should focus on selecting variables in a hypothesis-driven fashion instead of a data-driven fashion, and select variables which may be truly modifiable in a patient dataset. They could then explain how their tool could be used to make actual clinical recommendations. For example, a fruitful area in hypertension may be the sequence of drugs in hypertensive patients.

Answer: Thank you for your beneficial comment. Though we selected intervention variables in a data-driven manner, we agree with this comment that hypothesis-driven selection may be more useful in terms of practicality. In response to the reviewer’s comment, we performed an additional experiment on path planning with hypothesis-driven intervention variables (as described in the response to Comment (1)). In summary, four variables that would fluctuate with the clinical guideline–recommended treatments were selected by cardiologists as intervention variables, and the correspondence with guideline–recommended treatments is described in Supplementary Table 3. As a result, we demonstrated that our framework could provide actionable paths that corresponded to the clinicians' guidance.

Comment (6) Selection of reducing blood pressure was done by SBP above mean + (1-sigma). This does not reflect clinical guidelines, which are based upon cutoffs for hypertension (e.g. SBP >140mmHg). Can the authors justify their choice? It does not seem to be correct to select patients to modify in this fashion.

Answer: Thank you for your constructive comment on our paper. We agree with this comment that the selection of instances regarding reducing blood pressure should be based on cutoffs in clinical guidelines. In response to the reviewer’s comment, we performed path planning with the applicable instance selection cutoff changed to the standard value (SBP \geq 140 mmHg) of the Japanese hypertension guideline. We have modified the description in the main text as follows:

Before: “Assuming a scenario wherein the task is to lower the SBP in participants with higher values,

relevant instances were selected according to the following criteria: SBP above the mean + (1- σ), and no missing values in the intervention variables. The number of applicable instances was 391. ...planned paths were more actionable in 341/391 instances, and the median was 0.78.”

↓

After: “Assuming a scenario wherein the task is to lower the SBP in participants with higher values, relevant instances were selected according to the following criteria: predictive SBP ≥ 140 mmHg⁴², and no missing values in the intervention variables. The number of applicable instances was 259. ...planned paths were more actionable than baseline paths in 227/259 instances, and the median was 0.76.” (p. 8–9, 3rd paragraph in the subheading “Application of framework on SBP regression task” in the Results section)

We have also added the following reference.

Addition: “42. Umemura, S. *et al.* The Japanese Society of Hypertension Guidelines for the Management of Hypertension (JSH 2019). *Hypertension Research* **42**, 1235–1481 (2019).”

Along with the modification of the applicable instances in path planning, Figure 5d has also been modified. Since Figs. 6e–f were for an instance with predicted SBP < 140 , it has been changed to the result for an instance with predicted SBP ≥ 140 .

Minor

Comment (7) Reduced explanatory features to 25 with XGBoost recursive feature elimination.

Why was 25 selected? What criteria were used?

Answer: Thank you for pointing out the lack of description. In this study, we selected 25 as the number of features because feature reduction to 25 had little impact on the predictive scores in the validation data during recursive feature selection (RFE). To support this fact, the predictive scores at each stage of RFE have been added as Supplementary Fig. 4. Also, we have modified the description in the main text as follows:

Before: “RFE was performed with five-fold cross-validation split by participants, and the explanatory variables were reduced to 25”

↓

After: “RFE was performed with five-fold cross-validation split by participants, and the explanatory variables were reduced to 25, which showed little impact on predictive scores on validation data (Supplementary Fig. 4).” (p. 7, 1st paragraph in the subheading “Application on actual health checkup

dataset” in the Results section)

Comment (8) Explanatory variables related to SBP were excluded. It is unclear what the authors mean when they state this. Please provide more information. Why do the authors believe this is justified? It seems like these are precisely the variables which should be retained.

Answer: In this study, we intended to exclude diastolic blood pressure (DBP) and limb blood pressure values, which are almost synonymous with intervening in SBP. Our framework assumes that intervention in explanatory variables to improve response variables. Hence, situations like intervening in blood pressure (such as DBP) to lower SBP do not make much sense. To clarify the explanation, we have modified the description in the main text as follows:

Before: “We excluded measurement items related to blood pressure from the explanatory variables.”

↓

After: “We excluded diastolic blood pressure (DBP) and limb blood pressures from the explanatory variables for SBP prediction and creatinine for CKD risk prediction. Our framework assumes the intervention in explanatory variables to improve response variables. Hence, actions such as intervening in blood pressure (such as DBP) to lower SBP are not reasonable.” (p. 7, 1st paragraph in the subheading “Application on actual health checkup dataset” in the Results section)

Comment (9) Correlated features: Why were BMI and Waist size both selected? They should be highly correlated features and most feature-selection algorithms would remove one of them.

Answer: In response to the reviewer’s comment, we confirmed that the correlation coefficient between BMI and Waist was 0.85 (an attached file named “Correlartion_between_BMI_and_Waist.pptx”). Though the correlation is high, BMI is a height-corrected body composition parameter that is expected to affect predictive models differently than the waist. Also, the feature importance for each variable may be underestimated in tree-based algorithms when the highly correlated items exist (Kuhn M, Johnson K. 11.3 Recursive Feature Elimination. Feature Engineering and Selection: A Practical Approach for Predictive Models, Chapman and Hall/CRC; 2019). Therefore, both highly correlated items may remain in the RFE process. Because it was confirmed that the predictive score did not decrease in the item reduction to 25 in the RFE process as mentioned in response to Comment (7), the remaining of both items is considered to have a minor effect on the model performance.

Comment (10) Please provide a list of all variables considered at all steps in the process.

Answer: In response to the reviewer’s comment, we have added the detailed variable information to Supplementary Table 1 about (i) items that existed before RFE, (ii) items used for the prediction model construction, and (iii) items used for the stochastic model construction.

Comment (11) In 2020, having code available "upon reasonable request" for a mathematical model or tool is unacceptable. The authors should release their code for this method under appropriate license and make it available online for inspection.

Answer: In response to the reviewer’s comment, we have provided the code regarding our framework at https://github.com/clininfo/actionable_path_planning. Along with this, the Code availability section has been modified.

Comment (12) There is no such thing as "cured" hypertension (Table 1)

Answer: Thank you for your suggestive comment. We intended to use the word “cured” to refer to participants who had a medical history but were not currently receiving treatment. Based on the suggestions of the cardiologist, the description in Table 1 has been changed as follows:

Before: “Cured”

↓

After: “Past history” (p. 39, Table 1)

To Reviewer #2

Thank you very much for your helpful comments concerning our manuscript, which have helped us significantly improve the paper.

Remarks to the Author:

Advances in machine learning have meant that many machine learning models that produce impressive predictions/results are considered “black boxes,” as there is no easy way to explain how the models arrived at the results that they produced. This is especially problematic when such models are utilized in clinical medicine, where physicians often want to understand the “reasoning” that produced a particular result before acting on the result. The authors present a surrogate model based on hierarchical Bayesian models that can explain a nonlinear black box model and plan treatment or action paths.

Work on explainable AI is becoming more common in the biomedical domain and is often based on using decision trees as surrogate models that clinicians can more easily understand. A major strength of the current work is the use of an actual clinical dataset related to systolic blood pressure management. While the work presented is interesting, there are many missing details that would improve it.

Comment (1) Since the paper title states that it is presenting a health improvement framework, more examples related to health (as opposed to synthetic data) should be presented in the main paper (not in a supplement).

Answer: Thank you for your constructive comment on our paper. In response to the reviewer's comment, we conducted an additional experiment with the classification task for chronic kidney disease (CKD) risk as an example of applying the framework to actual health data, Iwaki Health Promotion Project (IHPP) dataset. The eGFR cutoff value for the CKD risk classification task was set to 60 mL/min/1.73m² based on the guidelines. To describe the results, we have added a section under the subheading "Application of framework on CKD risk classification task" to the Results section in the main manuscript (p. 10–11). In summary, based on the variables selected by RFE, a classification model with an AUC of 0.844 was constructed, followed by a stochastic surrogate model using hierarchical Bayesian modeling (Fig. 7a–c). Subsequently, we planned health improvement paths by changing the intervention variables selected in a data-driven and hypothesis-driven manner. It was demonstrated that the paths were planned through a node with a high probability (Fig. 7d, e), and the direction of change in intervention variables was consistent with conventional clinical knowledge (Fig.

8). Our framework has also been shown to be effective in interventions in instances with predictions of eGFR <60. Furthermore, as additional experiments, the framework was applied to the hypertension risk classification task and the eGFR regression task. These experimental results are described in the supplementary information due to the restriction on the number of characters, which demonstrates the effective path planning using our framework. From all the results given in the main manuscript and supplementary information, we believe that our framework is widely applicable to health data.

In the manuscript of the initial submission, the description of hierarchical Bayesian modeling only corresponded to the regression task. To describe the method on surrogate modeling for classification tasks, we have modified Fig. 2 and have added the following changes in the text:

Before: “We denote the measured explanatory values as \mathbf{x} and the predictions by the regression model as y . \mathbf{z} is the parameter of the mixture components, and k represents each mixture component. The data generative process is formulated as follows

$$\mathbf{x}_{cont} \sim N(\mathbf{m}_k, \Sigma_k) \quad (1)$$

$$\mathbf{x}_{disc} \sim \text{Categorical}(\boldsymbol{\phi}_{disc,k}) \quad (2)$$

$$y \sim N(\mu_k, \sigma) \quad (3)$$

$$k \sim \text{Categorical}(\boldsymbol{\pi}) \quad (4)$$

... according to the Cauchy distribution.”

↓

After: “We denote the measured explanatory values as \mathbf{x} and the predictions by the prediction model as y . \mathbf{z} is the parameter of the mixture components, and k represents each mixture component. The data generative process on a regression task is formulated as follows

$$k \sim \text{Categorical}(\boldsymbol{\pi}) \quad (1)$$

$$\mathbf{x}_{cont} \sim N(\mathbf{m}_k, \Sigma_k) \quad (2)$$

$$\mathbf{x}_{disc} \sim \text{Categorical}(\boldsymbol{\phi}_{disc,k}) \quad (3)$$

$$y \sim N(\mu_k, \sigma) \quad (4)$$

...according to the Cauchy distribution. In a classification task, y is a binary variable, and equations (4)–(7) are changed to the following formula accordingly:

$$y \sim \text{Bernoulli}(\theta_k) \quad (14)$$

$$\theta_k = \frac{1}{1 + e^{-\mu_k}} \quad (15)$$

$$\mu_k = \beta_{1,k} + \boldsymbol{\beta}_{2,k}^T \mathbf{x}_{cont} + \boldsymbol{\beta}_{3,k}^T \mathbf{x}_{disc} \quad (16)$$

$$\beta_{1,k} \sim N(0, 5) \quad (17)” \text{ (p.}$$

18–19, 1st paragraph in the subheading “Stochastic surrogate model with hierarchical Bayesian modeling” in the Methods section)

Comment (2) For each medical problem presented, the authors should make clear how many

features/predictors and gsamples were used in training the original nonlinear model and the surrogate, how missing data is handled in the original and in developing the surrogate and the rationale behind the methods chosen for handling missing data, and how well the original nonlinear model predicts the outcome of interest for both classification and regression problems.

Answer: Thank you for your careful comment. In response to the reviewer's comment, we have added details of the feature information used for each step of the framework to Supplementary Table 1. Also, we have added the description of the number of samples as Supplementary Table 2. As mentioned in the response to Comment (1), we performed a total of four medical applications (regression/classification tasks on SBP/eGFR) using our framework. For each experiment, (i) the handling method for the missing values and the reason for selection, and (ii) the predictive scores of the prediction models have been described. For example, in the SBP regression task in the section subheading "Application of framework on SBP regression task" in the Results section (p. 8–10), the method of handling missing values for the original nonlinear model is described in the sentence starting with "Following our framework, ..." (p. 8, 2nd paragraph). For the construction of the stochastic surrogate model, we used the data for which missing values were handled in the same method as for the original nonlinear model. We have made the following changes to clarify the description:

Before: "Subsequently, hierarchical Bayesian modeling was performed to construct the surrogate model."

↓

After: "Subsequently, hierarchical Bayesian modeling was performed to construct the stochastic surrogate model based on multiple imputed RFE-selected features and predicted SBP." (p. 8, 2nd paragraph in the subheading "Application of framework on SBP regression task" in the Results section)

For the systolic blood pressure regression task, the predictive score of the original nonlinear model is shown in Fig. 5b. Regarding the other three applications, corresponding location of sentences, figures, or tables for missing value completion and predictive score are shown below.

	Missing value completion	Predictive score
Classification task on SBP (Hypertension risk)	1 st paragraph in the subheading "Application of framework on hypertension risk classification task using IHPP dataset" in the Supplementary Information	Supplementary Table 7
Regression task on eGFR	1 st paragraph in the subheading "Application of framework on eGFR regression task using IHPP dataset" in the Supplementary	Supplementary Fig. 26

	Information	
Classification task on eGFR (CKD risk)	2 nd paragraph in the subheading "Application of framework on CKD risk classification task" in the Results section (p. 10)	Fig. 7b Supplementary Table 6

Comment (3) While a prospective study might be the best way to assess the effectiveness of treatment planning (path planning), an intermediate evaluation might involve asking several expert clinicians in the domain to assess the treatment plans produced by the surrogate model for their perceived utility.

Answer: Thank you for your constructive comment on our paper. In response to the reviewer's comment, we performed clinicians' evaluations on the health-improvement paths planned using our framework. We performed cardiologists' evaluation on SBP regression task and nephrologists' evaluation on CKD risk classification task. Regarding the SBP regression task, we have added 4th paragraph starting with "Second, we also planned paths with..." in the subheading "Application of framework on SBP regression task" section in the Results section (p. 9–10). Regarding the CKD risk classification task, results have been described in the 4th paragraph in the subheading "Application of framework on CKD risk classification task" section, which have been added in the response to Comment (1). To describe the methods, we have added a section under the subheading "Clinician assessment of health-improvement paths" to the Methods section in the main manuscript (p. 20–21). The summary of the evaluations is as follows.

We first compared the predicted value reductions between the framework-proposed paths and clinicians' interventions. Regarding the SBP regression task, five cardiologists who are board certified members of the Japanese Circulation Society participated in the evaluation. The evaluation was a blind test in which each cardiologist was asked to select the most suitable path from ten improvement paths: a framework-proposed path and nine randomly intervened paths (Supplementary Fig. 6). Here, the paths were constructed using four variables that the cardiologists pre-selected from the clinical guideline–recommended treatments as intervention variables: blood glucose, BMI, γ -GTP, and serum sodium. As a result of the test, the framework-proposed paths exhibited a significant decrease in predictive SBP compared to the paths selected by the cardiologists (Fig.5f). Similarly, the paths were evaluated by board certified nephrologists of the Japanese Society of Nephrology in the CKD risk classification task. Here, the paths were constructed using four variables that the nephrologists pre-selected from the clinical guideline–recommended treatments as intervention variables: triglyceride, RADIA, weight and Hb. The framework-proposed paths also exhibited a significant decrease in predictive CKD risk compared to the paths selected by the cardiologists (Fig.7f).

Subsequently, as the second evaluation, the utility of the paths was assessed by the clinicians.

The cardiologists and nephrologists assessed the practicality and informativeness of a framework-proposed path and a random path, respectively (Supplementary Fig. 7). As shown in Supplementary Table 4, the framework-proposed paths exhibited slightly (not significantly) lower practicality in the SBP regression task, whereas slightly higher in the CKD risk classification task, compared to the random paths. Also, the framework-proposed paths exhibited slightly (not significantly) higher informativeness in both the SBP regression task and CKD risk classification task. In positive opinions of the clinicians, for example, some clinicians evaluated that the framework-proposed health improvement paths according to the distribution of actual data were plausible and could lead to behavioral changes in patient. However, some comments were found that some improvements would be needed for practical use. For example, the values of intervention variables could fluctuate outside the range expected by the cardiologists. In fact, the current specifications of our framework performed path planning without setting upper and lower limits of the intervention variables to present objective health-improvement paths. In practical situations, it may be desirable for individual clinicians to set upper and lower limits for intervention variables. Detailed information on clinicians' evaluation of suggestive instances has been added as the subheading "Details of utility assessments of health improvement paths planned by framework" section in Supplementary Information.

These results would support the utility of the paths presented by our framework. We have made the following additions and changes to the Discussion section:

Addition: "As a result of clinicians' assessment, the paths planned using our framework exhibited significant improvement in predictive response variables compared to the paths selected by clinicians (Figs. 5f and 7f). Also, the improvement paths presented by our framework were informative for clinicians to some extent (Supplementary Table 4). However, the ratio of paths that clinicians evaluated as practical was not high (Supplementary Table 4). In this study, though we performed an objective path planning without considering clinical constraints, practical adjustments would be needed (detailed in the Supplementary Information)." (p. 12–13, 2nd paragraph in the Discussion section)

Before: "Additionally, when the target value of the response variable is determined based on guidelines or clinical knowledge, our framework can be applied by modifying the termination conditions of the search to reach the target value."

↓

After: "Additionally, there would be use cases where the target value of the response variable or intervention variable should be determined by clinicians (Supplementary Information). Our framework can be applied by modifying the termination conditions of the search to reach the target value." (p. 14, 4th paragraph in the Discussion section)

Before: “Although we evaluated that the change directions of intervention variables in some paths were consistent with clinical knowledge in the clinical application, it is necessary to verify the effectiveness of the paths through a prospective cohort study to suit the real-world applications of our framework.”

↓

After: “Also, we evaluated that the change directions of intervention variables in some paths were consistent with clinical knowledge in the clinical application. The clinician evaluation results suggested that while the framework would be promising, there are points to be adjusted for its practical application. It is necessary to verify the effectiveness of the paths through a prospective cohort study to suit the real-world applications of our framework.” (p. 15, 6th paragraph in the Discussion section)

Comment (4) As it currently stands, the paper makes claims about the potential utility of the methods presented for medical decision making but does not adequately substantiate these claims.

Answer: Thank you for your careful comment. In response to the reviewer’s comment, we conducted clinicians' utility assessments as mentioned in response to Comment (3). The added clinicians' evaluations indicate that the paths presented by our framework in clinical practice could have the potential to improve predictive response variables effectively and would be informative for clinicians.

Furthermore, to translate the output paths into actual clinical treatments, we added correspondence tables between the variables and clinical guideline–recommended treatments (Supplementary Table 3 and 5). For example, the altering γ -GTP was thought to correspond with alcohol restriction. If a path in the order of blood glucose reduction, γ -GTP reduction, and sodium reduction is proposed, it corresponds to intervention guidance in the order of dietary pattern change, alcohol restriction, and salt intake restriction according to the guideline–recommended treatments.

From these additions, we believe that our framework has a certain utility in medical decision making. However, we agree with the reviewer's comment that the claim was too strong. We have modified the expression as follows:

Before: “Our framework can provide clinicians with understandable and acceptable health improvement plans based on patient health data and given intervention variables. Accordingly, this is suitable for patient–clinician collaborative decision making on health interventions.”

After: “Our framework can provide clinicians with understandable and informative health improvement plans based on patient health data and given intervention variables. Accordingly, this

can be suitable for patient–clinician collaborative decision making on health interventions.” (p. 13, 2nd paragraph in the Discussion section)

Comment (5) It would help to cite more of the recent literature on explainable/intelligible models in medicine, such as:

Caruana R, Lou Y, Gehrke J, Koch P, Sturm M, Elhadad N. Intelligible models for healthcare: predicting pneumonia risk and hospital 30-day readmission. In: Proceedings of the 21st ACM SIGKDD international conference on knowledge discovery and data mining; 2015.

Lenert MC, Matheny ME, Walsh CG. Prognostic models will be victims of their own success, unless. J Am Med Inform Assoc. 2019;26(12):1645–50.

Zhang A, Teng L, Alterovitz G. An explainable machine learning platform for pyrazinamide resistance prediction and genetic feature identification of Mycobacterium tuberculosis. J Am Med Inform Assoc. 2020; November 20.

Answer: Thank you for your helpful comment regarding the various uses of explainable artificial intelligence (XAI) models in medicine. In response to the reviewer's comment, we have cited these literatures in the Introduction section as follows:

Before: “Machine learning (ML) technology has been extensively used in the medical field, especially for diagnosis support and disease prediction based on comprehensive patient information¹²⁻¹⁴. ...XAI is a research field on techniques that explain black-box ML predictions.”

↓

After: “Machine learning (ML) technology has been extensively used in the medical field, especially for diagnosis support and disease prediction based on comprehensive patient information¹²⁻¹⁵. ...XAI is a research field on techniques that explain black-box ML predictions, and it has been applied to the medical ML models where interpretability is often required^{18,19}.” (p. 3, 2nd paragraph in the Introduction section)

We have also added the following references.

Addition:

“15. Lenert, M. C., Matheny, M. E. & Walsh, C. G. Prognostic models will be victims of their own success, unless... *Journal of the American Medical Informatics Association* **26**, 1645–1650 (2019).”

“18. Caruana, R. *et al.* Intelligible Models for HealthCare: Predicting Pneumonia Risk and Hospital 30-day Readmission. *Proceedings of the 21th ACM SIGKDD International Conference on Knowledge Discovery and Data Mining - KDD '15* 1721–1730 (2015).”

“19. Zhang, A., Teng, L. & Alterovitz, G. An explainable machine learning platform for pyrazinamide resistance prediction and genetic feature identification of *Mycobacterium tuberculosis*. *Journal of the American Medical Informatics Association* **0**, 1–8 (2020).”

To Reviewer #3

Thank you very much for your thoughtful comments concerning our manuscript, which have helped us significantly improve the paper.

Remarks to the Author:

Since the intention (at least, how I perceived it) is to use the proposed framework to make actionable decision in clinical practice, more rigorous experiments are needed. These include:

Comment (1) more extensive simulations using realistic data generating models. Even though simulations such as the one presented in the paper are useful to get intuition about the framework, more realistic data scenarios need to be created. For instance, the diagonal covariance matrices (Supplementary Figure 2) are unrealistic in real-life applications. Most, if not all datasets, exhibit some correlation between the explanatory variables. Also, the dimensionality of the simulations need to be higher than 3.

Answer: Thank you for your constructive comment on our paper. In response to the reviewer's comment, we evaluated our framework using more realistic dataset. We have added the results as a section subheading "Validation of framework on 5D synthetic dataset" in the Supplementary Information. In summary, we generated a five-dimensional dataset in which correlations between variables exist (Supplementary Fig. 40). We built nonlinear ML models, followed by a stochastic surrogate model using hierarchical Bayesian modeling (Supplementary Fig. 12a–f). Subsequently, we planned paths to decrease predicted response variable values. It was demonstrated that the paths were planned through a node with a high probability (Supplementary Figs. 12g–i and 13–15). From these results, our framework applied to the dataset with covariance and could be used to plan paths with five intervention variables. Since path planning with a larger number of intervention variables is difficult due to computational costs, we have added the investigation to clarify the trade-off between the number of intervention variables and the number of steps. This detail is described in the response for Comment (3). For specific and individual cases and attributes, heuristics may be efficient, which is a future task.

Comment (2) investigation of the impact of misspecification of the Bayesian surrogate model. The normality assumption for the covariates (eq 1) is not very realistic. For instance, blood pressure does not have the real line as support. Extensive simulations under different data generating models can provide indication of the robustness of the surrogate model.

Answer: Thank you for your insightful comment. As the reviewer mentioned in the comment, we used a distribution based on the normal distribution for continuous variables (eq. 2 [eq. 1 in the initial submission manuscript]). On the other hand, the stochastic surrogate model was based on hierarchical Bayesian modeling that assumed a mixture of normal distributions (eq. 1), allowing representation of flexible data distribution more than a normal distribution. Successful path planning with a more complex dataset, where a normal distribution does not fit, in an additional experiment performed in response to Comment (1) supports the robustness of stochastic surrogate models. As the reviewer mentioned, it may be better to assume a mixture distribution of distributions other than the normal distribution, for example, when the sample size is small. The stochastic surrogate model can indeed express the probability for the range of each variable that deviates from the distribution of the actual data. When the sample size is sufficient, the range of variables with high nodal probabilities can be estimated appropriately, and the nodal probabilities in such a deviate range are low. Hence, it is considered that there is little effect on our path planning in which nodes with high probability are mainly searched.

Comment (3) one or two more real-life applications related to medicine. The authors need to consider examples with more intervention variables as these are the rule rather than the exception in clinical applications. This will allow to showcase how the search algorithm performs in such settings. (Since the algorithm seems to be making local moves I have some doubts whether it performs well in higher dimensions.)

Answer: Thank you for your beneficial comment on our paper. In response to the reviewer's comment, we performed three additional experiments.

First, we conducted the path planning with intervention variables selected by the hypothesis-driven manner according to the guidelines in addition to the data-driven manner based on feature importance. The results of this additional experiment have been added as the 4th paragraph starting with the sentence “Second, we also planned paths with...” in the subheading “Application of framework on SBP regression task” in the Results section (p. 9–10). In summary, our framework was capable of planning effective health improvement paths in other intervention variable set (Fig. 5e).

Second, we performed an additional experiment with the chronic kidney disease (CKD) risk classification task as an example of applying the framework to actual health data, Iwaki Health Promotion Project (IHPP) dataset. We have added a section under the subheading "Application of framework on CKD risk classification task" to the Results section in the main manuscript (p. 10–11). As other additional experiments in the Supplementary Information, the framework was also applied to the hypertension risk classification task and the eGFR regression task. As detailed in the response to Reviewer #2's Comment (1), these results show the successful path planning in various scenarios

using our framework.

Third, though the sentence starting with "From the perspective of expanding the proposed framework, ..." in the Discussion section (p. 14, 5th paragraph) stated that the calculation cost would increase as the intervention variables increased, a detailed experiment was conducted on this point. Regarding the SBP regression task and CKD risk classification task described in the main manuscript, path planning was performed by increasing the intervention variables to ten variables, which could be intervened, selected from the top of feature importance (Fig. 5a and 7a). The number of steps of the path decreased compared to the result for five intervention variables with the same iteration count ($L = 20,000$) (Supplementary Fig. 30). The calculation time was about 150 min/instance. These results are considered to be due to the increase in the number of nodes that exist within a certain number of steps in the multi-dimensional grid graph. As the number of intervention variables, that is, the number of dimensions in the grid graph, increases, the number of nodes within a certain number of steps exponentially increases (Supplementary Fig. 31). Though we took an exact solution to ensure optimal path planning, heuristic search methods could be promising in high dimensional path planning. Considering the actual application in medicine, it is not realistic to instruct the patient to improve many items, such as 10 or 15 items. Clinicians' evaluations conducted in response to Reviewer #2's Comment (3) show that health-improvement paths with four intervention variables were informative for clinicians (Supplementary Table 4). Hence, it would be sufficient to plan a health improvement path using around five variables as intervention variables in medical applications. Regarding the calculation cost with increased intervention variables, we have modified the description in the main text as follows:

Before: "Subject to our experimental conditions, approximately 10 min were required for path planning per instance."

↓

After: "Subject to our experimental conditions ($L = 20,000$ and five intervention variables), approximately 10 min were required for path planning per instance. Under the condition of more intervention variables, the number of steps of the planned optimal paths decreased, and the calculation time became longer even in the same iteration count, L (Supplementary Fig. 30–31)." (p. 14, 5th paragraph in the Discussion section)

Comment (4) The disadvantage of the surrogate model. The authors claim that the Bayesian surrogate model allows to calculate the probability of counterfactual values (Results section, 4th paragraph). This statement needs justification. It is a very strong statement given the method uses observation data. To draw causal conclusions from observational data additional assumptions need to be made (see Chapter 3, Hernán MA, Robins JM, 2020. Causal Inference:

What If.). All the surrogate model does is estimate the joint densities of the explanatory variables. If some values are not seen in the dataset due to biases arising from the observational nature of the data, then they will be assigned low posterior probability and the path planning optimisation will avoid these regions of the space.

Answer: Thank you for your insightful comment. We agree with this comment that all the surrogate model does is to estimate the joint probability densities of the variables and not to calculate the probability of counterfactual values. We apologize that we included the misleading expression "counterfactual value" in the text. By estimating the causal effect of each intervention variable on the response variable through more controlled studies related to individual applications, it may be possible to construct a stochastic surrogate model considering the causality and to present health-improvement paths that are more consistent with the clinician's consideration. We have changed the representation and have added the limitation regarding causality as follows:

Before: "Note that this stochastic surrogate model represents a probability density on not only a given dataset but also virtually changed values, i.e., counterfactual values."

↓

After: "Note that this stochastic surrogate model represents the probability density for all possible states of variables." (p. 5, 3rd paragraph in the subheading "Path planning framework using surrogate Bayesian model" in the Results section)

Before: "...we could calculate the probability of counterfactual values. In our framework, the optimal path was defined as a sequence of the counterfactual values with high probability in the surrogate model."

↓

After: "...we could calculate how easy it was to take the state of the combination of variables. The framework output the most actionable (optimal) path for a state that improved the prediction value, which was a sequence of the changed values with high probability in the surrogate model (detailed in the methods section)." (p. 6, 4th paragraph in the subheading "Path planning framework using surrogate Bayesian model" in the Results section)

Before: "The explanatory variables for setting the counterfactual values were called intervention variables in this study."

↓

After: "The explanatory variables for setting the virtually changed values were called intervention variables in this study." (p. 6, 5th paragraph in the subheading "Path planning framework using

surrogate Bayesian model” in the Results section)

Addition: “Also, because of the limitation of the observational study data, the stochastic surrogate model only calculated the joint probability density of a set of variables and did not explicitly express the intervention effect considering the causality⁶⁰. By estimating the causal effect of each intervention variable on the response variable through more controlled studies related to individual applications, it may be possible to construct a stochastic surrogate model considering the causality and to present health-improvement paths that are more consistent with the clinician's consideration.” (p. 15, 6th paragraph in the Discussion section)

We have also added the following reference.

Addition: “60. Robins M. James, M. A. H. *Causal Inference - what if*. Boca Raton: Chapman & Hall/CRC (2020).”

Comment (5) **Actionability is not formally defined in any part of the paper? The only reference close to a definition is “From the computational perspective, we used the negative logarithm of actionability, defined as the product of node probabilities on a path, as a cost of the path” which raises further questions such as how the “cost of the path” is defined/evaluated?**

Answer: Thank you for your beneficial comment. In this study, we defined a probability of the node as the probability of taking the node calculated using the surrogate model. The actionability of the path was defined as the product of nodal probabilities on a specified path. To clarify the definition of the actionability, we have changed the descriptions as follows:

Before: “Furthermore, the actionability was defined as the product of node probabilities on a specified path.” (p. 6, 5th paragraph in the subheading “Path planning framework using surrogate Bayesian model” in the Results section)

↓

After: “The actionability of the path was defined as the product of probabilities of taking a series of variable states on a specified path (detailed in the methods section).” (p. 5, 1st paragraph in the subheading “Path planning framework using surrogate Bayesian model” in the Results section)

and

“We defined a probability of the node as the probability of taking the node calculated using the

surrogate model. Furthermore, the actionability was defined as the product of nodal probabilities on a specified path.” (p. 19, 1st paragraph in the subheading “Path planning using stochastic surrogate model” in the Methods section)

The purpose of our path search algorithm was to obtain the path with the maximum actionability to each node in the search area. In the actual path search calculation, the path was planned as an optimization problem to minimize the negative logarithm of the actionability to each node. This is because the actionability is the product of the probabilities of a series of nodes, which is hard to handle directly in computation. In the sentence the reviewer mentioned, we intended to use the negative logarithm of the actionability as the "cost of the path" to minimize. To clarify the representation, we have made the following modification:

Before: “From the computational perspective, we used the negative logarithm of actionability, defined as the product of node probabilities on a path, as a cost of the path.”

↓

After: “From the computational perspective, we used the negative logarithm of actionability as a path cost to minimize, which is mathematically synonymous with maximizing actionability.” (p. 20, 1st paragraph in the subheading “Path planning using stochastic surrogate model” in the Methods section)

Furthermore, we have clarified the cost initialization process in the pseudocode (lines 2–3 in Fig. 3).

Comment (6) In the introduction the authors refer to other similar ML approaches arising from the field of XAI. It will be useful to outline the differences (advantages and disadvantages) of the field compared to the proposed framework and maybe discuss further in the Discussion section. For example, since both LIME and SHAP use a surrogate model the main contribution of the proposed methodology seems to be the path planning algorithm. If that is the case, can we use the surrogate model from LIME or SHAP followed by the proposed path planning optimisation routine? If yes, then the proposed method should be evaluated compared to LIME/SHAP or other similar approaches. If no, mention why.

Answer: The scope of LIME and SHAP is to explain the prediction reason for the ML model for each instance, which is realized by constructing a locally applicable surrogate model. The local surrogate models of these methods are used to present the predictive contribution of the explanatory variables used in the ML models for each instance. However, the important nature of our *stochastic* surrogate model is to be able to calculate the probability of taking the state of certain variable sets, which is not feasible with the conventional surrogate model. This nature of the stochastic surrogate model enables

actionable path planning in our framework. We have achieved the construction of the *stochastic* surrogate model using hierarchical Bayesian modeling. Therefore, it is not possible to search the paths in our framework using the surrogate model of LIME or SHAP. To clarify this point, we have made the following changes:

Addition: “Unlike the conventional surrogate model, the stochastic surrogate model based on the hierarchical Bayesian model enables the calculation of the probability of being the state of a given variable set.” (p. 4, 3rd paragraph in the Introduction section)

Before: “Conventional XAI methods, such as LIME and SHAP, cannot provide concrete improvement paths” (p. 12, 1st paragraph in the Discussion section)

↓

After: “Though conventional surrogate models applied in XAI methods, such as LIME and SHAP, are useful for identifying individual factors that contribute to prediction, they cannot provide the probability of taking variable states. In our framework, the construction of a stochastic surrogate model based on hierarchical Bayesian modeling enabled the estimation of joint probability densities for virtually changed variables and actionable path planning.” (p. 12, 1st paragraph in the Discussion section)

Comment (7) About the saerch algorithm. The breadth-first search algorithm needs to be briefly explained or at least offer a reference for the reader who wants to know more.

Answer: Thank you for your thoughtful comment. In response to the reviewer’s comment, we have added the following reference.

Addition: “27. Skiena, S. S. *The Algorithm Design Manual*. Springer Science+Business Media (Springer London, 2008). doi:10.1007/978-1-84800-070-4.”

Comment (8) In addition, I have a couple of reservations on the implementation. The authors state “this algorithm was used to obtain the most actionable path to the node that achieved the most improved predictive value within the search iteration count, L”. This raises the issue that if I run the algorithm for L + 1 iterations I may come up with a different answer. This seems undesirable. Some form of convergence to an optimum needs to be established. Otherwise, there is no notion of an “optimal health-improvement treatment path” (Results, section, 4th paragraph).

Answer: Thank you for your careful comment. In the initial submission manuscript, the explanation about the optimal path might be unclear. Our path search algorithm plans the most actionable path, i.e., the optimal path, to reach every node in the range of search iteration count. After that, the optimal path to the node with the most improved predicted value in the range of the search iteration count is provided as the output of the algorithm. Therefore, if $L + I$ is set to the number of iterations, though the search range will be expanded, the optimal paths to the nodes in the range of search iteration count L remain unchanged. To clarify the definition of “optimal path”, we have made following changes:

Before: “In our framework, the optimal path was defined as a sequence of the counterfactual values with high probability in the surrogate model.”

↓

After: “The framework output the most actionable (optimal) path for a state that improves the prediction value, which was a sequence of the changed values with high probability in the surrogate model (detailed in the methods section).” (p. 6, 4th paragraph in the subheading “Path planning framework using surrogate Bayesian model” in the Results section)

Before: “The purpose of this algorithm was used to obtain the most actionable path to the node that achieved the most improved predictive value within the search iteration count, L .”

↓

After: “The purpose of this algorithm was used to obtain the most actionable (optimal) path for each node. The output path was the optimal path to the node with most improved predictive value within the search iteration count, L .” (p. 19–20, 1st paragraph in the subheading “Path planning using stochastic surrogate model” in the Methods section)

Comment (9) Also, what happens if the algorithm gets stuck in a node? Under normal circumstances the algorithm could be re-initialised from different values. But, here the initial node is fixed since it corresponds to the observed covariate values of a given patient.

Answer: As the reviewer mentioned in the comment, the initial node in the path search is fixed because it corresponds to the observed values of a given patient. In this study, the output of the path search algorithm is the optimal path destined for the node with the most improved prediction in a range of iteration count specified by the user. The search efficiency would be worse when the distribution for the intervention variable is such as a uniform distribution or distribution with large variance. Even in such a case, the search algorithm does not stop in a node. Of course, since the path search is performed within a specified iteration count due to the constraint of calculation cost, the node with the most improved predicted value in the entire variable space is not always reached by the search. In medical

applications, since extreme fluctuations in intervention variables are considered to be unrealistic, it is reasonable to present the optimal health-improvement paths to the node within a stochastically close range, i.e., a certain number of iteration counts.

Comment (10) Lastly, the presentation of the pseudocode needs more work: what are the `currentnode.cost`, `negihbor.node.neg.log.prob` etc? I did not find a definition in the entire document. These quantities are crucial for someone who wants to implement the algorithm, so they should be clearly defined.

Answer: Thank you for your careful comment. In response to the reviewer’s comment, we added more detailed description on the pseudocode (Fig. 3). In summary, initialization of the cost and visited state are inserted in lines 2–5. Details about the cost are as described in response to Comment (5). Regarding "neg_log_prob", we supplemented that it is a negative logarithm of node probability (line 10). Also, the following description has been added to the text:

Addition: “In lines 2–5, the cost and visited state (the optimal path was searched or not) of nodes were initialized.” (p. 20, 2nd paragraph in the subheading “Path planning using stochastic surrogate model” in the Methods section)

Furthermore, we have provided the code regarding our framework at https://github.com/clininfo/actionable_path_planning. Along with this, the Code availability section has been modified.

Comment (11) Sensitivity analysis. The authors should evaluate the sensitivity of the results to a number of choices. First, what are implications if a different ML regression model is used? Even though XGBoost is fine, the authors need to show whether the path planning changes under different models. This is because different models lead to different predictions which may have (big) impact to all further downstream analysis.

Answer: Thank you for your beneficial comment on our paper. In response to the reviewer’s comment, we conducted additional experiments applying our framework to the ML model other than XGBoost: random forest (RF) and support vector machine (SVM). The results of the performance of the prediction models, hierarchical Bayesian modeling, and path planning are summarized in the Figures/Tables described in the table shown below.

	XGBoost	RF	SVM
3D synthetic dataset	Supplementary Fig. 3 Fig. 4	Supplementary Fig. 9 Supplementary Fig. 10	Supplementary Fig. 9 Supplementary Fig. 10
5D synthetic dataset	Supplementary Fig. 12 Supplementary Fig. 13	Supplementary Fig. 12 Supplementary Fig. 14	Supplementary Fig. 12 Supplementary Fig. 15
Regression task on SBP	Fig. 5 Fig. 6	Supplementary Fig. 16 Supplementary Fig. 17	Supplementary Fig. 16 Supplementary Fig. 18
Classification task on SBP (Hypertension risk)	Supplementary Table 7 Supplementary Fig. 22 Supplementary Fig. 23	Supplementary Table 7 Supplementary Fig. 22 Supplementary Fig. 24	Supplementary Table 7 Supplementary Fig. 22 Supplementary Fig. 25
Regression task on eGFR	Supplementary Fig. 26 Supplementary Fig. 27	Supplementary Fig. 26 Supplementary Fig. 28	Supplementary Fig. 26 Supplementary Fig. 29
Classification task on eGFR (CKD risk)	Supplementary Table 6 Fig. 7 Fig. 8	Supplementary Table 6 Supplementary Fig. 19 Supplementary Fig. 20	Supplementary Table 6 Supplementary Fig. 19 Supplementary Fig. 21

In summary, similar paths were planned for the same instances regardless of applied ML algorithms in the synthetic datasets (Fig. 4 and Supplementary Figs. 9–15). On the other hand, the paths differed in the same instances depending on the ML algorithms in the actual dataset (Figs. 6, 8, Supplementary Figs. 16–29). This may be because the prediction model properties for the complicated actual differed between the ML algorithms. Our framework is a method to interpret for a specified ML model like other XAI methods. Our framework presents the optimal path to the node with the most improved predictions of a particular prediction model within the search iteration count, L . Therefore, the different prediction model properties may cause a change in the set of explanatory variables that improve prediction, which results in a change in the destination node itself and the optimal path accordingly. Regarding the applications of the framework to predictive models other than XGBoost, we have added the following description:

Addition: “The supplementary information provides the results of applying the framework to random forest and support vector machine (Supplementary Figs. 9–29 and Supplementary Table 6–7). In the synthetic datasets, similar paths were planned for the same instances regardless of applied ML algorithms (Fig. 4 and Supplementary Figs. 9–15). However, in the actual dataset, the paths differed in the same instances depending on the ML algorithms (Figs. 6, 8, and Supplementary Figs. 16–29). The actual dataset was more complicated than the synthetic datasets, which resulted in the construction of the prediction models with different properties. Our framework aimed to provide an optimal path to the destination node that would improve the predictions of the ML model. Therefore, the destination

node itself was changed when the properties of the ML model were different, which resulted in large differences in the planned paths.” (p. 13, 3rd paragraph in the Discussion section)

Comment (12) Second, please evaluate the sensitivity of the results on the choice of priors in the surrogate model and motivate their the choice.

Answer: Thank you for your insightful comment. We conducted simple experiments in the early study and selected stable priors. In response to the reviewer’s comment, we performed detailed experiments on the sensitivity analysis regarding the hyperparameters of the priors. We utilized weakly informative prior distributions and noninformative prior distributions since we assumed no prior information was given in our experiments. Using 3D and 5D synthetic datasets, we performed hierarchical Bayesian modeling with changed hyperparameters as shown in Supplementary Figs. 33a and 34a. As a result, it was found that similar WBIC values were obtained in the changes of prior distributions examined (Supplementary Figs. 33b and 34b). These results suggest that our prior distribution settings work well in situations where prior knowledge is not assumed. When using our framework as an actual application, it may be effective to set priors that reflected the prior knowledge in the field. We have added the following text about the selection of priors:

Addition: “The priors were selected from noninformative prior distributions or weakly informative prior distributions. The relationships between prior distribution hyperparameters and WBIC are shown in Supplementary Figs. 33–34.” (p. 19, 3rd paragraph in the subheading “Stochastic surrogate model with hierarchical Bayesian modeling” in the Methods section)

Comment (13) Third, discuss/motivate the choice for the range of mixture components, why 1-8?

Answer: Thank you for your careful comment. When the number of mixture components increased, the sampling number would increase in the Markov chain Monte Carlo algorithm. Because this caused a longer calculation time, we had to set an upper limit for the number of mixture components. In response to the reviewer’s comment, we conducted additional experiments regarding the relationship between the number of mixture components and planned paths. In the normal setting in this study, though we used the stochastic surrogate model of the mixture components with the lowest WBIC for path planning, we planned paths using the stochastic surrogate model of different mixture components in this experiment. In both the synthetic datasets and the actual health dataset, the planned paths were almost unchanged under a superabundant number of mixture components (Supplementary Fig. 35–38). Therefore, it is reasonable to set an upper limit on the number of mixture components to reduce

computational costs. We have explained the choice for the range 1–8 of the number of mixture components by modifying the description as follows:

Before: “We set 1–8 as the range of mixture components.”

↓

After: “Furthermore, we set 1–8 as the range of mixture components from the viewpoint of calculation costs and supplementary experimental results (Supplementary Figs. 35–38). The supplemental results show that the planned paths almost unchanged under a superabundant number of mixture components.”

(p. 19, 3rd paragraph in the subheading “Stochastic surrogate model with hierarchical Bayesian modeling” in the Methods section)

Reviewers' Comments:

Reviewer #2:

Remarks to the Author:

In the revised manuscript, my major concerns with the previous version of the manuscript were addressed by (1) the addition of an application of the authors' treatment/path planning framework to chronic kidney disease, and (2) the inclusion of five board certified cardiologists' and five board certified nephrologists' assessments of proposed treatment plans/paths for systolic blood pressure management and CKD risk classification. While the clinician assessment results indicate that there would be additional steps required to make the authors' proposed framework practical and informative for clinical use, the study overall is an interesting and valuable contribution.

It may be interesting for follow-on work (beyond the scope of the current paper) to assess why more nephrologists seemed to evaluate the optimal path plan as useful (practical or informative) compared to cardiologists.

Reviewer #3:

Remarks to the Author:

The authors are to be congratulated for the additional amount of impressive work both in terms of quality and quantity. They have addressed all my comments. But a couple more concerns arose in the process:

1. Comment (11) Sensitivity of the results to the choice of the ML model.

In addition to XGBoost the authors have used two more ML models to evaluate the sensitivity of the output (ie optimal paths) to the choice of the model. The results for the real data are given in Figs. 6, 8, and Supplementary Figs. 16–29. The figures (corresponding to the real data) show that the paths derived under the three ML models are different.

This implies that if I go to two different doctors/clinics or hospitals that use the proposed framework but different ML models I will be recommended different treatments. To me, this is highly undesirable. This also makes me question the notion of "optimality" of the proposed paths. If I want to go from point A to point B I define the shortest route as the optimal one. This notion is not applicable in this paper. For instance, in the CKD risk classification task for instance 6 (Fig 8e, Supplementary Figure 20e, and Supplementary Figure 21e) I always start at the same initial values and finish at almost identical predicted probabilities but whilst following completely different paths.

Would a possible solution be to use the observed y values when fitting the surrogate model instead of the predicted ones? This may render the surrogate model and subsequently the path search less dependent on the ML model. Would this be possible? What do the authors think?

2. Another comment related to Fig 6f, and Supplementary Fig 18f. In both figures, the path does not seem to go through or terminate on high-density regions. This is what the authors claim their algorithm is doing. Please comment.

3. Comment (12) Second, please evaluate the sensitivity of the results on the choice of priors in the surrogate model and motivate their choice.

The authors have included several additional priors for the surrogate model and show that the

WBIC remains similar across the different settings (Supplementary Figs. 33b and 34b). But what about the paths chosen under the different priors? I don't understand how similar WBIC values imply similar paths? In addition, I was expecting the results to be presented for the real data. I believe this would be more meaningful. My question is: Do small changes in the priors of the surrogate model (for a given ML model) lead to small or large changes in the chosen optimal for the real data?

To conclude, the authors present an interesting idea. The modifications to the manuscript have made it more accessible. Nevertheless, I don't see such an idea being implemented in practice any time soon. This is not an issue per se given that the authors seem to target presenting the method rather than creating a tool that can be utilized. From this perspective it is noteworthy. But I believe (and the authors mention in the discussion) considerable further research is needed before such a tool can be transformed into something useful and "actionable".

Reviewer #4:

Remarks to the Author:

Thanks for the detailed replies; the revision addresses all of the reviewer's comments.

Reviewers' comments to Author & Reply to reviewers

To Reviewers

We would like to thank the Reviewers for having taken in reviewing our manuscript thoughtfully.

To Reviewer #2

In the revised manuscript, my major concerns with the previous version of the manuscript were addressed by (1) the addition of an application of the authors' treatment/path planning framework to chronic kidney disease, and (2) the inclusion of five board certified cardiologists' and five board certified nephrologists' assessments of proposed treatment plans/paths for systolic blood pressure management and CKD risk classification. While the clinician assessment results indicate that there would be additional steps required to make the authors' proposed framework practical and informative for clinical use, the study overall is an interesting and valuable contribution.

It may be interesting for follow-on work (beyond the scope of the current paper) to assess why more nephrologists seemed to evaluate the optimal path plan as useful (practical or informative) compared to cardiologists.

Thank you for your meaningful comment on our manuscript. In future works, we would like to focus on the more practical aspects of our framework.

To Reviewer #3

Thank you very much for your insightful comments regarding our manuscript.

The authors are to be congratulated for the additional amount of impressive work both in terms of quality and quantity. They have addressed all my comments. But a couple more concerns arose in the process:

1. Comment (11) Sensitivity of the results to the choice of the ML model.

In addition to XGBoost the authors have used two more ML models to evaluate the sensitivity of the

output (ie optimal paths) to the choice of the model. The results for the real data are given in Figs. 6, 8, and Supplementary Figs. 16–29. The figures (corresponding to the real data) show that the paths derived under the three ML models are different.

This implies that if I go to two different doctors/clinics or hospitals that use the proposed framework but different ML models I will be recommended different treatments. To me, this is highly undesirable. This also makes me question the notion of “optimality” of the proposed paths. If I want to go from point A to point B I define the shortest route as the optimal one. This notion is not applicable in this paper. For instance, in the CKD risk classification task for instance 6 (Fig 8e, Supplementary Figure 20e, and Supplementary Figure 21e) I always start at the same initial values and finish at almost identical predicted probabilities but whilst following completely different paths.

Would a possible solution be to use the observed y values when fitting the surrogate model instead of the predicted ones? This may render the surrogate model and subsequently the path search less dependent on the ML model. Would this be possible? What do the authors think?

Answer: Thank you for your constructive comment. The direct answer to the question of whether observed y can be used for the surrogate modeling, which the Reviewer mentioned in the last paragraph in the comment, is yes. It is possible to fit the surrogate model using observed y in principle. Certainly, this method would reduce the dependence of the constructed surrogate model on the ML model. However, the ML model to predict y corresponding to virtually changed X from the initial values would be important because predictive y is used for probability calculation in path planning in our framework. Therefore, we considered that it would be necessary to build a surrogate model that fits the predicted y from the given ML model.

The most ML model-independent, extreme way to plan the same path for the given initial values is considered to be the one that does not use the ML model itself. For example, this would be a method in which only observed data points, which consist of X and y , are connected to plan the optimal path for given initial values. However, there are two major problems with this method. The first is the need for a large amount of observational data that cover the search area. Second, path planning would be difficult when the initial values deviate from the observed data distribution because of biased real-world data. Especially in medical applications, because it is practically difficult to obtain necessary and sufficient observation data due to data bias depending on the region, environment, and disease area, these problems are expected to be particularly large. Therefore, we have achieved to plan the path via X for which the corresponding y is not obtained as the measured value by combining the ML model and the stochastic surrogate model. A possible future direction for planning a more robust path to the given initial values could be to build a stochastic model with high predictive performance and

perform both prediction and path planning based on this one model. We have modified the description in the main text as follows:

Before: “The framework output the most actionable (optimal) path for a state that improved the prediction value, which was a sequence of the changed values with high probability in the surrogate model (detailed in the methods section).”

After: “The framework output the most actionable (optimal) path for a state that improved the prediction value, which was a sequence of the changed values with high probability in the surrogate model for the given ML model (detailed in the methods section).” (p. 6, 4th paragraph in the subheading “Path planning framework using surrogate Bayesian model” in the Results section)

2. Another comment related to Fig 6f, and Supplementary Fig 18f. In both figures, the path does not seem to go through or terminate on high-density regions. This is what the authors claim their algorithm is doing. Please comment.

Answer: Thank you for your insightful comment. As described in the Methods section, the purpose of our path planning algorithm is to obtain the most actionable (optimal) path for each node. The output path is the optimal path to the node with the most improved predictive value within the search iteration count L . That is, the destination node would be selected without considering the destination nodal probability. Even though the nodes with low probabilities are selected as the destination nodes in the examples the Reviewer mentioned, our framework estimates the most actionable paths from the initial to terminal nodes. In real clinical situations, destination nodes with such low probabilities are assumed to deviate from normal values. To avoid this problem, by slightly changing the path search condition to constrain the search range in intervention variables based on clinicians' judgment and clinical practice, the destination nodes with low probability could be avoided in our framework (as described in the 4th paragraph in the Discussion section).

3. Comment (12) Second, please evaluate the sensitivity of the results on the choice of priors in the surrogate model and motivate their choice.

The authors have included several additional priors for the surrogate model and show that the WBIC remains similar across the different settings (Supplementary Figs. 33b and 34b). But what about the paths chosen under the different priors? I don't understand how similar WBIC values imply similar paths? In addition, I was expecting the results to the presented for the real data. I believe this would

be more meaningful. My question is: Do small changes in the priors of the surrogate model (for a given ML model) lead to small or large changes in the chosen optimal for the real data?

Answer: Thank you for your constructive comment on our paper. In response to the Reviewer’s comment, we examined the output paths when the priors were changed for the real data. We performed stochastic surrogate modeling with the different priors on the XGBoost models for the SBP regression task and CKD risk classification task (Supplementary Figs. 35a and 36a). In most cases, similar WBIC values to our model settings in the main text were obtained in the changes of prior distributions examined (Supplementary Figs. 35b and 36b). Subsequently, we planned paths using the stochastic surrogate model of the mixture components with the lowest WBIC for each prior distribution. The planned paths for the six instances described in the main text (instance 1–3 for SBP regression task and instance 4–6 for CKD risk classification task) are shown in Supplementary Figs. 35c–e and 36c–e. In some cases, different paths would be planned with specific hyperparameters, e.g., hyperparameter setting 5 in instance 4 (Supplementary Fig. 36c). However, in most cases, similar paths with our model settings in the main text were obtained (although slight changes in the order of improvement could be observed in some cases). Though specific hyperparameter settings yielded a surrogate model with a better WBIC value in the SBP regression task (hyperparameter setting 10 in Supplementary Fig. 35a), the planned paths were similar to our model settings in the main text (Supplementary Figs. 35c–e). These results suggest that our prior distribution settings also work well in the real data. We have added Supplementary Figs. 35–36 and the following text:

Before: “The relationships between prior distribution hyperparameters and WBIC are shown in Supplementary Figs. 33–34.”

After: “The relationships between prior distribution hyperparameters and WBIC are shown in Supplementary Figs. 33–36.” (p. 19, 3rd paragraph in the subheading “Stochastic surrogate model with hierarchical Bayesian modeling” in the Methods section)

To conclude, the authors present an interesting idea. The modifications to the manuscript have made it more accessible. Nevertheless, I don’t see such an idea being implemented in practice any time soon. This is not an issue per se given that the authors seem to target presenting the method rather than creating a tool that can be utilized. From this perspective it is noteworthy. But I believe (and the authors mention in the discussion) considerable further research is needed before such a tool can be transformed into something useful and “actionable”.

Thank you again for your thoughtful comment on our paper. We agree with the Reviewer's comment that further improvements are needed for the practical use of the proposed framework. In future research, we will examine and modify our framework from a more practical point of view.

To Reviewer #4

Thanks for the detailed replies; the revision addresses all of the reviewer's comments.

We appreciate the time and effort you have dedicated to reviewing our manuscript.

Reviewers' Comments:

Reviewer #3:

Remarks to the Author:

Thanks for the replies; the revision addresses my comments.